

# Recommendations for processing atmospheric attenuated backscatter profiles from Vaisala CL31 Ceilometers

Simone Kotthaus[1], Ewan O'Connor[1,2], Christoph Münkel[3], Cristina Charlton-Perez[4],

Andrew M. Gabey[1], and C. Sue B. Grimmond[1]

[1]Department of Meteorology, University of Reading, Reading, RG6 6BB, UK

[2]Finnish Meteorological Institute, Helsinki, 00101, Finland

[3]Vaisala GMBH, Hamburg, 22607, Germany

[4]Met Office, Meteorology Building, University of Reading, Reading, RG6 6BB, UK

*Correspondence to*: Simone Kotthaus (s.kotthaus@reading.ac.uk)

**Abstract.** Ceilometer lidars are used for cloud base height detection, to probe aerosol layers in the atmosphere (e.g.

detection of elevated layers of Saharan dust or volcanic ash) and to examine boundary layer dynamics. Sensor optics and acquisition algorithms can strongly influence the observed attenuated backscatter profiles; therefore, physical interpretation of the profiles can require corrections to be applied carefully. This study addresses the commonly deployed Vaisala CL31 ceilometers. Attenuated backscatter profiles are studied to evaluate the impact of both the generation of the hardware and version of the firmware. In response to this work and discussion within the CL31/TOPROF user community (TOPROF is a

European COST Action aiming to harmonise remote sensing networks across Europe), Vaisala released new firmware (versions 1.72 & 2.03) for the CL31 sensors. These firmware versions are tested against previous versions showing that several artificial features introduced by the data processing have been removed. Hence, it is recommended to use this recent firmware if attenuated backscatter is to be analysed. To allow for consistent processing of historic data, correction procedures are developed that account for the range-dependent electronic background signal and other artefacts detected in

data collected with older firmware. Recommendations are made for the processing of attenuated backscatter observed with Vaisala CL31 sensors, including the estimation of noise which is not provided in the standard CL31 output. Taking these aspects into account, attenuated backscatter profiles from Vaisala CL31 ceilometers are considered to provide valuable information for a range of applications including, for example, atmospheric boundary layer studies, detection of elevated aerosol layers, and model verification.



# 1. Introduction

Ceilometer lidars are widely used to characterise clouds (Illingworth et al., 2007), providing reliable estimates of cloud base height with the ability to detect multiple cloud layers (Martucci et al., 2010). They were originally developed as 'cloud base recorders', but their attenuated backscatter profiles can also provide information on rainfall (Rogers et al., 1997), formation and clearance of fog (Haeffelin et al., 2010), drizzle properties (when combined with cloud radar; O'Connor et al., 2005) and for the study of aerosols, including elevated layers of Saharan dust (Knippertz and Stuut, 2014), biomass burning (Mielonen et al., 2013) or volcanic ash (Nemuc et al., 2014), and particles dispersed within in the atmospheric boundary layer (Tsaknakis et al., 2011). Using aerosols as a tracer, boundary layer dynamics, including mixing height and the formation of residual layers, can be inferred from ceilometer attenuated backscatter observations (Münkel et al., 2006).

Although ceilometers are regarded as the most basic automatic lidar (Emeis, 2010), Wiegner et al. (2014) conclude that they can be used to detect the location and extent of aerosol layers and to derive the aerosol backscatter coefficient if the instrument is carefully calibrated. Such observations are highly valuable for the evaluation of numerical weather prediction (NWP) and air-quality models (Emeis et al., 2011) and are increasingly used in forecast verification. Several national meteorological services (NMS) and research centres are currently evaluating the potential of using ceilometer profile observations for data assimilation (Illingworth et al., 2015).

Ceilometers, often referred to as ALC (automatic lidars and ceilometers), observe attenuated atmospheric backscatter profiles and usually perform sophisticated cloud base height detection algorithms. As they can operate automatically for long time periods without maintenance or human intervention even in extreme climates (Bromwich et al., 2012), they are widely deployed operationally by NMS (e.g. www.dwd.de/ceilomap) and long-term research campaigns (e.g. www.met.reading.ac.uk/micromet). International initiatives, such as the E-Profile programme (http://www.eumetnet.eu/e-profile) of the Network of European Meteorological Services (EUMETNET), aim to facilitate the exchange of observational data by harmonising the ALC network across Europe. The European COST Action TOPROF (http://www.toprof.imaa.cnr.it/) works in close collaboration with E-Profile to ensure that the necessary steps of quality control are incorporated into the common data processing (Illingworth et al., 2015). As ceilometers are manufactured by several companies, the sensor optics, hardware components and software algorithms may differ significantly. Discussions in the TOPROF community have revealed the importance of a detailed understanding of instrument specifics to identify the necessary processing steps that enable appropriate interpretation and harmonisation of the final data products.

This study documents the important processing steps that should be considered when analysing attenuated backscatter profiles from Vaisala CL31 ceilometers. Observations from two ALC networks (Sect. 2) are used to illustrate relevant data processing aspects grouped into (i) those addressing the whole vertical profile of the observed signal (Sect. 3) and (ii) those



specific to the lowest range gates (Sect. 4). Depending on the firmware version, the CL31 instrument internal processing may introduce certain artefacts that should be accounted for if the attenuated backscatter is required for analysis. It is shown how the signal strength can be used for quality assurance (Sect. 5) and findings are summarised in the form of recommendations for the processing of CL31 profile observations (Sect. 6).

## 2. Instrument description

The Vaisala CL31 ceilometer transmits a very short pulse of 110 ns (corresponding to an effective pulse length of about 16.5 m, e.g. Weitkamp, 2005). The receiver uses an avalanche photo diode (APD) detector to process the returned signal. The instrument oversamples the backscattered signal at a temporal rate which corresponds to the range resolution setting. The reported range $r$ (i.e. distance from the instrument) is the centre of a range gate. Low pass filtering by the instrument extends and shapes the pulse response so that different vertical resolutions can be achieved depending on the sample rate. For example, a sample rate of 15 MHz is required to achieve a range resolution of 10 m, where the first observation reported at 10 m is backscattered signal for 5-15 m from the ceilometer. Every 2 s, $2^{14}$ laser pulses are emitted with a frequency of 10 kHz which takes about 1.64 s. After this period there is an idle time of 0.36 s used to perform the cloud base detection algorithm before the next set of $2^{14}$ laser pulses is emitted. After a certain number of gates have been sampled, the firmware slightly changes operation mode; thus, regions of increased noise are introduced into the backscatter profiles at two ranges: 4940 m and 7000 m. Samples collected during the 2 s intervals are averaged over certain internal intervals to create the raw signal $\tilde{P}$ at a rate defined by the reporting interval selected by the user (2 – 30 s). The internal averaging interval is specific to the firmware.

The spectral wavelength of the laser diode used in the Vaisala CL31 is $905 \pm 10$ nm at 298 K, as stated by the laser manufacturer. Vaisala finds the uncertainty of the nominal centre wavelength to be well below 10 nm. Typical spectral width (Full Width at Half Maximum, FWHM) is 4 nm. Lasers produced from the same wafer agree in terms of the centre wavelength, however, the exact centre wavelength is unknown to the user. For a specific laser the centre wavelength is slightly temperature-dependent (0.3 nm K$^{-1}$). The CL31 system heater near the laser transmitter serves to stabilise the laser temperature in cold environments. Further, both window transmission and laser pulse energy can have an impact on the attenuated backscatter signal. The laser heat sink temperature, window transmission and laser pulse energy are therefore monitored and reported continuously. Status information (i.e. diagnostics, warnings and alarms) is included in the data message which helps to identify whether maintenance is required (e.g. window needs cleaning, transmitter is failing).

The detector of the CL31 ceilometer responds to the laser pulse backscattered of from molecules, aerosols, rain drops and both liquid and ice cloud particles. It also responds to noise originating from both external (e.g. daytime solar radiation) and internal (e.g. electronic) sources. The hardware-related noise is larger than the Rayleigh signal associated with clear air so





that the latter is too small to be distinguished. Vaisala states that the variance of the electronic noise signal is range-independent. Solar insolation increases the current through the APD, but as the amplifiers are AC-coupled, the relatively slowly varying solar signal (almost DC) does not get to the A/D converter (the AC-coupling time constant is 1 ms). This filtering results in a variable zero-level (i.e. noise has negative and positive values) in the form of a voltage offset that

accounts for temporal variations in the noise. While the AC coupling removes the low frequency signal from varying solar radiation, the latter still increases signal noise (shot noise in APD due to DC current). For short data acquisition intervals, backscatter values can be below zero. Electronic noise is also a function of system properties (e.g. detector temperature, transmitter lens area; Gregorio et al., 2007; Vande Hey, 2014) and can therefore be analysed by the manufacturer prior to field deployment. Heaters achieve partial laser and detector system thermal stabilisation in cool or cold conditions.

The Vaisala CL31 firmware has changed with certain developments in the hardware, i.e. the receiver (CLR) and engine board (CLE) where the internal processing takes place. These updates have resulted in the creation of a range of firmware versions. For CLE311 + CLR311, the firmware versions 1.xx are used, while sensors with CLE321 + CLR321 run firmware versions 2.xx. Changing the ceilometer transmitter CLT generation is not connected to a change in firmware. The internal

averaging interval differs slightly with firmware version (Table 1). In Vaisala CL31 firmware versions below 1.72 and versions 2.01, and 2.02, the internal averaging interval is set to 16 s for range gates below 2400 m if the reporting interval is greater than 8 s. For reporting intervals between 5 – 8 s, the internal averaging interval is set to 8 s below 1800 m and 16 s between 1800 – 2400 m, respectively. For reporting intervals below 5 s internal averaging below 1200 m is 4 s; only for the minimum reporting interval of 2 s, internal averaging is set to 2 s below 600 m. Above 2400 m, the internal averaging is 30 s

for all reporting intervals. In firmware 1.72 and 2.03, the internal averaging interval is 30 s for the entire profile and does not change with reporting interval (Table 1). If a reporting interval is selected that is shorter than the internal averaging interval, consecutive profiles overlap in time and are hence not completely independent.

Observations from two ceilometer networks (Table 2) are used in this study to illustrate aspects of the data acquisition and

processing of Vaisala CL31 ceilometers. The London Urban Micromet Observatory (LUMO; www.met.reading.ac.uk/micromet) is a measurement network collecting observations of many atmospheric fields to investigate climate conditions within and around Greater London, UK (for interactive map see http://www.met.reading.ac.uk/micromet/LUMA/Network.html). The Met Office operates an ALC network (http://www.metoffice.gov.uk/public/lidarnet/lcbr-network.html) across the UK with different manufacturers/models,

including Vaisala CL31 ceilometers. Four sensors of the LUMO network in central London and one Met Office sensor located 60 km west of central London are used here.

Long-term observations are available from four CL31 ceilometers with different generations of hardware and various firmware versions. Over time, the LUMO network firmware versions have changed from the first LUMO sensor deployed in





2006 with version 1.56 (Table 2). Sensors A and B are the old hardware generation with the CLE311 board, as is the Met Office sensor W, while LUMO sensors C and D have engine boards CLE321. For both sensors A and B, the transmitter has been upgraded from CLT311 to CLT321 during their operation. While the LUMO sensors are set to acquire data every 15 s with a vertical resolution of 10 m, data from the Met Office ceilometer have a resolution of 30 s and 20 m.

## 3. Profile Corrections

### 3.1. Background correction

While the amount of backscattered signal depends on the distance of the source to the instrument, the sources of noise should, in principle, be range-independent. However, given that the time-dependence of the data acquisition is linked to the spatial domain, some range-dependence may be found in the noise, i.e. the signal *background* $P^{bg}$. As CL31 measurement design accounts for temporal variations in the background noise by introducing a variable zero-level (Sect. 2), the 'raw' signal is inherently partly background-corrected. Thus, only the range-dependent component of the background $P^{bg}(r)$ needs to be accounted for to derive the entirely background-corrected signal $\hat{P}$. $\tilde{P}^{raw}$ denotes the range-corrected values reported by the ceilometer and $P^{raw}$ is the signal after reverting the range correction (see Sect. 3.2).

The background $P^{bg}(r)$ can combine effects of the electronic noise and those associated with internal processing by the instrument as some Vaisala CL31 firmware versions slightly alter the partly background-corrected signal. For some firmware versions the signal is shifted artificially so that the background noise of the respective data is biased and is no longer centred on zero (e.g. for data collected with version 1.71 there are more negative than positive values). This procedure is applied in the processing to improve the detection of cloud base height as it amplifies the difference between the signal backscattered from cloud droplets and atmospheric scatterers. Increasing this difference also facilitates visual interpretation of clouds based on the backscattered signal. Hereafter, this bias is referred to as 'cosmetic shift' $P^{cs}$. Thus, to derive the entirely background-corrected signal from CL31 output the complete background $P^B$, composed of range-dependent, instrument related background $P^{bg}(r)$ and the cosmetic shift, needs to be accounted for:

$$\hat{P}(r) = P^{raw}(r) - P^B(r) = P^{raw}(r) - P^{bg}(r) - P^{cs}(r). \tag{1}$$

For data collected with firmware 1.72 or 2.03, no cosmetic shift is incorporated ($P^{cs}(r) = 0$) so that the complete background correction is associated with the range-dependent electronic background ($P^B(r) = P^{bg}(r)$). The impact of firmware version on the partly background-corrected signal with cosmetic shift is illustrated using observations from different clear-sky days (no elevated dust, aerosol layers or cirrus) recorded with CL31 sensors using a range of firmware versions (Fig. 1a). Under such conditions it is assumed that the only source of atmospheric signal above the atmospheric boundary layer (ABL) is very weak molecular scattering. Molecular scattering depends on the known air molecule density profile, which varies very little, so that, while the absolute values of the profile depend on the intervening optical properties



(extinction), the gradient of the profile in the absence of any scattering particles should show no variation. In practice, molecular scattering at the instrument wavelength is very weak, typically below the sensitivity of the instrument, so that the profile is dominated by the hardware-generated noise. Therefore, no systematic differences in the shape of the observed profiles would be expected and obvious departures from the anticipated shape in the profiles can be attributed to the data

acquisition and processing. A suitable method to identify shape discrepancies is to create signal-range histograms (Fig. 1a) using 24 hours (or more) of data. The most obvious effect revealed by the range histograms is a step-change in the width of the distributions at 2400 m evident for all versions apart from 1.72 and 2.03. This step-change is introduced by the averaging of the sampled signal that is applied internally by the instrument's firmware (Sect. 2, Table 1). The decreased averaging for range gates < 2400 m performed for earlier firmware increases the signal noise (see Sect. 5.2). Data acquired with version

1.72 or 2.03 are more consistent across the range gates as the whole profile is treated equally with an internal averaging interval of 30 s.

The range histograms (Fig. 1a) show the impact of the incomplete background correction, i.e. that electronic background and cosmetic shift are not (entirely) accounted for. The complete background depends on the firmware version and is a function

of range with both range-dependent electronic background and cosmetic shift causing a systematic pattern in the observed profiles. The cosmetic shift is particularly strong for version 1.71. To capture both effects, profiles are analysed during times when atmospheric variations are expected to be small and instrument conditions are stable (Fig. 2). Average $P^{raw}$ profiles are extracted when noise induced by solar insolation is absent (four hours around midnight), when cloud cover is low (< 10% of the hour), no fog is present, the window transmission is reasonable (> 80% on average), laser pulse energy is high (> 98%

of nominal energy), and sufficient data are available (> 90% of the hour). Only range gates > 2400 m are analysed to avoid the impact of changing internal averaging intervals (Sect. 2) at this critical range (Fig. 1a) and to minimise the signal from the ABL (unlikely to extend above 2400 m over London around midnight). Median vertical profiles (with inter-quartile range (IQR) shading) are displayed for common setup conditions (Fig. 2, on the right), i.e. grouped by the combinations of sensor, firmware and transmitter (CLT), respectively.

The climatology of night-time profiles (Fig. 2) reveals a small temporal variability with a seasonal cycle (amplitude ~ 50%) that indicates a temperature dependence of the electronic background. Several features appear distinct in the spatial domain (Fig. 2) at certain range gates. For all sensors and firmware versions, a discontinuity is evident at around both 5000 m and 7000 m. These regions of increased noise are introduced by the data storage procedure (Sect. 2). Another pattern observed to

be systematic with range is a wave-like structure superimposed over the random noise for ceilometer B operating with transmitter CLT321 (Fig. 2b) and ceilometer C in general (Fig. 2c). Such patterns are introduced by a physical 'ringing' of the detector which overlays a vertically alternating positive and negative bias on top of the signal noise. While this wave-type bias tends to be similar for successive profiles (regions with positive and negative amplitude overlap), it is not constant over long time periods (i.e. the same range gate may have a positive or negative bias at different times). Hence, if sufficient





profiles are averaged, the ringing becomes less apparent in the climatology. For example, comparing median profiles of sensor B (Fig. 2b) operating with firmware 1.61 (number of profiles N = 222) and 1.71 (N=497) shows that averaging over a long time period decreases the wave-structure. As seen from the presented comparison, this ringing is sensor-specific (e.g. a higher frequency is detected for sensor B compared to sensor C, Fig. 2b, c). While ringing may occur for ceilometers with

both the CLE311 (Fig. 2b) and CLE321 engine boards (Fig. 2c), of the ceilometers tested only sensors operating a transmitter of type CLT321 were found to have the ringing effect. The firmware version does not affect this wave-type bias as it is solely a hardware-related, electronic contribution to the background $P^{bg}(r)$.

Assuming the actual information content related to atmospheric backscatter is low in the selected night-time profiles (i.e.

above the ABL the signal contribution is small compared to the noise in the absence of clouds), the median climatology grouped by firmware plus transmitter configuration (Fig. 2, right) describes the systematic, range-dependent electronic background plus cosmetic shift (i.e. $P^{bg}(r) + P^{cs}(r)$). Although the range of values is large, IQR and median profiles have rather consistent statistics: the shape of the background profile depends mainly on the firmware used. This becomes particularly evident when comparing the median night-time climatology profiles for the various configurations directly (Fig.

3). The seasonality evident in the time series (Fig. 2, left) is related to the laser heat sink temperature which is used to further classify the background profiles into three sub-classes (Fig. 3a, see legend).

To evaluate whether the night-time climatology is a suitable basis on which to determine the background signal profile, 'dark-current' measurements for four LUMO sensors with recent firmware and hardware configurations (Fig. 3b) are used.

The ceilometer window is covered by a Vaisala *termination hood* to mimic full attenuation (i.e. supposedly no signal backscattered to the receiver). Only internal contributions (e.g. the electronic background noise) should be in the recorded signal. To eliminate any transient behaviour in the lowest range gates (not shown) the hood measurements were taken for 30 min periods. Profiles from the same laser heat sink temperature classes are selected for comparison. For most sensors and firmware, the median night-time climatology agrees very well with the profile observed by the termination hood

measurement (Fig. 3b). Only for ceilometer A (firmware 1.71) does the termination hood measurement have a slightly different shape, albeit with a similar order of magnitude. Thus, dark current reference measurements give confidence that the night-time climatology profiles provide reasonable background profiles and are not significantly influenced by backscatter from atmospheric particles. This finding is extremely useful as it allows for background noise profiles of ceilometer sites that were operated in the past or that are difficult to access (e.g. termination hood measurements are unfeasible) to be evaluated

based on the observed profile data alone.

The median background profiles from the same firmware tend to agree qualitatively apart from version 1.72 for which profiles of sensors A & B (both with the same configuration or engine board (CLE311), receiver (CLR311), and transmitter (CLT321)) are of opposite sign (Fig. 2a, b and Fig. 3a). This behaviour is likely associated with the noise generated by the





specific hardware components rather than the firmware. The step change when profiles change sign is also evident in the climatology of the night-time profiles (time series in Fig. 2). The complete background profile $P^B$ ($=P^{bg}(r) + P^{cs}(r)$) is positive ($\sim$ 2-5 $\cdot$ 10$^{-14}$ arbitrary units (a.u.) at 2400 m) below 7000 m for firmware 1.61, generally decreasing with altitude range. For all background profiles observed with firmware 1.71, the strong, negative bias ($\sim$ 12-14 $\cdot$ 10$^{-14}$ a.u. at 2400 m)

associated with the cosmetic shift applied, causes overall negative background profiles which increase (i.e. absolute values decrease) slightly with range below about 5500 m. Background profiles from newer hardware (sensors C and D, Fig. 2c, d) have a similar shape independent of firmware (i.e. 2.01, 2.02, 2.03). While the shape of the median profiles is roughly comparable for the two sensors, the negative bias below 4600 m is stronger for ceilometer D than C. For these newer sensors (C and D, Fig. 2; Fig. 3a) the magnitude of the electronic background decreases with increasing temperature.

Vaisala state (firmware release note) that no deliberate cosmetic shift is implemented in versions 1.72 and 2.03. Given the similarity of background profiles from the earlier release versions, the cosmetic shift in versions 1.56, 1.61, 2.01, and 2.02 is also negligible and the complete background $P^B(r)$ is only generated by the range-dependent electronic background $P^{bg}(r)$. Only firmware 1.71 profiles are shifted significantly towards negative values. The long-term estimates of instrument and

firmware specific background (electronic effects and cosmetic shift; Fig. 2, right) are used as the correction profiles $P^B_{night}$ derived for the 2400 – 7700 m range:

$$P^B_{night}(r) = \; [P^{bg}(r) + P^{cs}(r)]_{night}. \tag{2}$$

Below 2400 m the night-time climatology is unsuitable to calculate background noise profiles as aerosols and humidity in the ABL attenuate the signal. Unfortunately, termination hood measurements are unusable for derivation of a correction profile given the extremely high values observed below 1000 m for all CL31 ceilometers tested (Fig. 3b). As no information

is available about the shape of the background profile below 2400 m, values for the background are assumed to be constant in height up to this range. The night-time profiles present a suitable, temporally constant background correction for sensors without significant cosmetic shift.

Data with cosmetic shift (i.e. those collected with firmware 1.71) show strong diurnal variations in the background signal in

response to the noise zero-level, which is shifted dynamically during data acquisition (Sect. 2). Because this is done internally by the firmware, the exact zero-level is not available for post-processing use. However, it can be approximated by the average signal $Z(t)$ across the top range gates where the atmospheric contribution to the signal is negligible. The calculation of $Z(t)$ is the same as used when estimating the noise-floor $F(t)$ (see Eq. (13), Sect. 5.2). While $Z(t)$ is usually small with no distinct diurnal pattern, it is clearly affected by the response of the zero-level to background solar radiation for

data collected with firmware version 1.71. The night-time background profiles (Eq. (2)) determine the range-dependence of the background correction, while its magnitude further depends on $Z(t)$. The nocturnal average (4 h around midnight) of the signal at the top of the profile $\bar{Z}_{night}$ is subtracted so that the amplitude of $Z(t)$ only introduces diurnal variations and the





background correction $P^B(r)$ remains close to the climatology $P^B_{night}(r)$ when solar radiation is absent. For firmware with no significant cosmetic shift, the background correction $P^B(r)$, describing the electronic background, can be determined by the night-time profiles. For firmware version 1.71 with strong cosmetic shift, profiles are offset by a diurnal pattern in the shape of the estimated zero-level:

$$P^B(r) = \begin{cases} P^B_{night}(r) - (Z(t) - \bar{Z}_{night}), & firmware\ version = 1.71 \\ P^B_{night}(r), & firmware\ version \neq 1.71 \end{cases} \tag{3}$$

The derived background correction $P^B(r)$ can be applied in the post-processing to estimate the entirely background-corrected signal $\hat{P}$ without effects of cosmetic shift from the data recorded (Eq. (1)). This correction reduces the range-dependence of the observed signal so the range-histograms of $\hat{P}$ (not shown) are more symmetric around zero than those of $P^{raw}$ (Fig. 1a) in all range gates in the free atmosphere.

For firmware versions without cosmetic shift or where cosmetic shift is negligible, the background profile consists solely of the range-dependent electronic background which is small (on average $< |5 \cdot 10^{-14}|$ a.u.). Implications of these instrument specific variations for observations within clouds or in the ABL, where backscatter values tend to be large and mostly positive, might be limited. However, the positive electronic background of sensors with older hardware (CLE311) can dominate signal differences expected at the top of the ABL at times and the cosmetic shift in version 1.71 clearly affects

observations within the ABL (Sect. 5.2). Note that the cosmetic shift and electronic background should be carefully evaluated before using noise for quality control purposes (see Sect. 5).

### 3.2.    Range correction

For a given concentration of atmospheric scatterers (cloud, aerosol, molecules) the strength of the backscattered signal returned to the ceilometer telescope and detector decreases by the square of the range $r$. Therefore, to relate scattering

coefficients at different ranges, the raw signal $P^{raw}$ or the background-corrected signal $\hat{P}$ is multiplied by $r^2$ at each range gate to obtain the range-corrected signal:

$$\tilde{P}(r) = \hat{P}(r) \cdot r^2. \tag{4}$$

Vaisala instruments have an option where the range correction is applied only to the signal in the lower part of the profile up to a set range $r_{H2}$, where it is implicitly assumed that most of the data at further ranges consists of noise (setting: '*Message profile noise_h2 off*'). If no clouds are present in the profile, the raw signal is multiplied by a constant, height-invariant scale

factor $k_{H2}$ above $r_{H2}$ (CL31: $r_{H2} = 2400$ m and $k_{H2} = r^2_{H2} = 2400^2$). The partly range corrected signal $\tilde{P}^{raw}_{H2}$ has two segments:

$$\tilde{P}^{raw}_{H2} = \begin{cases} P^{raw}(r) \cdot k_{H2}, & r > r_{H2} \\ P^{raw}(r) \cdot r^2, & r \leq r_{H2} \end{cases} \tag{5}$$



When clouds are detected, the cloud signal is range-corrected using Eq. (4), for range gates where cloud is determined to exist. To create a fully range-corrected signal from such observations for the whole vertical profile (according to Eq. (4), i.e. as if run with the setting '*Message profile noise_h2 on*') in the absence of clouds, the scale factor needs to be reversed and the range correction applied to the observations above $r_{H2}$:

$$\tilde{P}^{raw} = \begin{cases} \tilde{P}_{H2}^{raw}(r)/k_{H2} \cdot r^2, & r > r_{H2} \\ \tilde{P}_{H2}^{raw}(r), & r \leq r_{H2} \end{cases} \quad (6)$$

For ceilometers operating with '*Message profile noise_h2 on*', all firmware applies the range correction throughout the entire profile and no constant scale factor is incorporated in this processing step. Hence it is recommended to operate with this setting turned on.

The background-corrected signal $\hat{P}^{raw}$ can then be derived by reversing the range correction (Eq. (4)). The range-histograms
of the range-corrected signal (Fig. 1b, c) illustrate the increase in signal variability with range. After applying the full range correction (Eq. (6)) to observations from a CL31 operated with '*Message profile noise_h2 off*' (rightmost panel in Fig. 1), the variability of the background corrected signal is height-invariant above the ABL while the expected increase is found in the range corrected signal (Fig. 1c).

## 4.  Low-level corrections

Although Vaisala suggests that the attenuated backscatter profile is reliable down to the first range gate, Sokół et al. (2014) document a distinct local minimum in CL31 attenuated backscatter observations at the 5[th] range gate persisting throughout their whole observational campaign. As others have found artefacts in CL31 profiles below 70 m (e.g. Martucci et al. 2010; Tsaknakis et al. 2011) these lowest layers are often excluded during processing. If looking for mixing layer height or top of the ABL, including these lowest layers could result in false detection of internal boundary layers. Artefacts in the lowest
80 m could be related to the incomplete optical overlap (Sect. 4.1) but are more likely associated with a low-level correction introduced by Vaisala to prevent unrealistically high values in the near range when the window is obstructed  and a hardware-related perturbation (Sect. 4.2).

### 4.1.  Optical overlap

The receiver field of view reaches complete optical overlap with the emitted laser beam at a certain distance above the
instrument. This overlap depends on instrument design. Overlap correction functions can be applied to partly account for this effect. They are dimensionless multiplication factors (0 (nearest the instrument) to 1 (range gates above the point of complete overlap)) which are determined empirically (e.g. Campbell et al., 2002). The overlap correction may either be performed by firmware or during post-processing. Uncertainty remains for observations at the closest range gates (e.g. Martucci et al., 2010; Sokół et al., 2014; Vande Hey, 2014).





Applying an optical overlap correction $O(r)$ to the entirely background-corrected signal, yields the overlap corrected signal:

$$\hat{P}^{OC}(r) = \hat{P}(r) \Big/ O(r) \qquad (7)$$

Vaisala ceilometers have a single-lens, coaxial beam setup. For the CL31, complete optical overlap is reached at about 70 m from the instrument (Fig. 4) and an overlap correction is performed by the firmware. Vaisala overlap functions are verified

both by ray tracing simulations and laboratory measurements.

### 4.2.   Obstruction correction

Given the primary function of cloud base height detection, Vaisala designed CL31 firmware to identify and address effects causing extremely high backscatter values outside of clouds. Under severe window obstruction (e.g. leaf on the window), values for the first range gates would be unrealistically high. A correction is applied to restrict the backscatter profile in the

lowest ranges. At times, this correction introduces extremely small values at ranges < 50 m that are clearly offset from the observations above this height. In addition to the artefacts from the obstruction correction, for some sensors, backscatter values in the range of 50-80 m are slightly offset by a hardware-related perturbation. Both the artefacts from the obstruction correction and the hardware-related perturbation do not impact the detection of clouds, vertical visibility or boundary layer structures (above 80 m). Only if internal boundary layers are to be analysed below 80 m, is their impact required to be

accounted for. The issues are not firmware-specific, apart from versions 1.72 and 2.03 in which the artefacts of obstruction-correction and hardware-related pertubation have been removed. Hence, these low-level artefacts are expected to be consistent in time when analysing data from older firmware.

To evaluate the effect of the obstruction-correction and hardware-related perturbation, profiles of the overlap-corrected

signal in the lowest 100 m are normalised by the value at 100 m ($\tilde{P}^{OC}(n)/\tilde{P}^{OC}(10)$, with n = range gate). Analysis of daytime (11-16 UTC) median profiles for selected conditions ($\tilde{P}^{OC} < 200 \times 10\text{-}8$ a.u.; no absolute calibration applied) in the lowest 400 m over a year (2013) (Fig. 5a-c) show that the normalised profiles have a consistent shape across the four LUMO CL31 sensors (Fig. 5a). A small reduction in backscatter is detected at 80 m (8th gate), a distinct peak at 50 m (5th gate) and rather similar values in the lowest four gates (< 40 m). The artefacts are of smaller magnitude in observations from sensor B (Table

2). The ratio of the values across the lowest four range gates to the value observed at the 10th range gate appear to have two different regimes. While for most profiles the normalised overlap- and range-corrected signal ranges between 1.0 and 1.2 (Fig. 5b), a small fraction of samples is marked by consistently lower values ($\tilde{P}^{OC}(2)/\tilde{P}^{OC}(10) < 0.8$; Fig. 5c). This effect is likely explained as an artefact of the obstruction correction while the deviations at 50 m and 80 m are associated with the hardware-related perturbation. The observed range provides uncertainty information for the detection of the low-level

artefacts; the ratio reaches values of 40-50% at the 5th range gate and commonly < 20% for the remaining range gates below



100 m. Low-level profiles observed with firmware 1.72 or 2.03 (Fig. 5d) show the artefact of obstruction correction and hardware-related perturbation were removed with these updates.

The general shapes of normalised profiles are consistent in time (i.e. peak always found at 5[th] range gate), while absolute magnitudes vary slightly when backscatter values exceed those used to calculate the climatology (Fig. 5). To parameterise the temporal variations of the artefacts, relations between normalised values at different range gates are used. The 2013 median daily profiles (Fig. 5a-c) were used to establish linear relations describing any observation in the 6[th]-9[th] range gate as a function of the observation in the 5[th] range gate (Table 3). Generally, the first four gates have height-invariant factors so that the values at the 1[st], 2[nd] and 4[th] gate can be expressed as a function of the normalised value at the 3[rd] range gate. These correction functions can help to reduce the processing artefact due to the obstruction correction and the hardware-related offset (Fig. 6). Although some residual effects may remain, extreme vertical gradients encountered within the lowest 100 m of the original range corrected signal observed by the CL31 ceilometers are mostly removed. A similar correction can be applied to rainy periods (not shown). In fog, the effect of the perturbations are negligible as the extinction caused by the fog droplets is comparatively stronger by several orders of magnitude.

## 5. Absolute backscatter and quality assurance

### 5.1. Absolute calibration

The attenuated backscatter $\beta$ describes the range-corrected, entirely background-corrected and overlap-corrected signal calibrated by the lidar constant $C$.

$$\beta = \tilde{P}^{OC}/C \tag{8}$$

The range correction can be reversed for the attenuated backscatter to yield the entirely background-corrected and overlap-corrected attenuated backscatter $\hat{\beta}$:

$$\hat{\beta} = \beta/r^2. \tag{9}$$

The lidar constant $C$ is a function of the range-independent parameters of the lidar equation, including the speed of light, the area of the receiver telescope, the temporal length of a laser pulse, a system efficiency term and the mean laser power per pulse (Weitkamp, 2005). It depends on the instrument receiver design and its laser. When the instrument is new, system efficiency and laser power are high. At this stage, the lidar constant for internal calibration is determined by a factory-based test ($C = C_{factory}$). Even with regular cleaning and maintenance, the performance of a sensor changes over time (e.g. aging of the laser, changes in window transmissivity). To account for such possible variations in laser output and detector capability over time, the ceilometer firmware monitors the laser output energy and determines a relative calibration-correction factor $c_{monitor}(t)$ which is a time-specific lidar constant applied internally:


$$C_{internal}(t) = C_{factory} \cdot c_{monitor}(t). \tag{10}$$

Over time, this internal checking of the instrument performance potentially provides a continuous relative calibration. Given that the signal output by the ceilometer already has the internal calibration applied, it is labelled 'attenuated backscatter' by the manufacturer. However, it has been shown that the internal calibration factor $C_{internal}$ does not always fully represent the actual lidar constant (e.g. O'Connor et al. 2004) and that an absolute calibration should be performed in sufficiently known

atmospheric conditions. Given the background noise of the CL31 sensors dominates over the molecular backscatter (Sect. 2), the stratocumulus cloud technique (O'Connor et al., 2004) is the most appropriate calibration technique for the Vaisala sensors. This agrees with the findings of the TOPROF community (Maxime Hervo, Meteo Swiss, personal communication). The stratocumulus cloud technique relates the observed signal to the known integrated attenuated backscatter coefficient associated with thick liquid clouds. This absolute calibration technique is applied externally, i.e. as part of the post-

processing:

$$\beta = \left( \tilde{P}_{internal}^{OC} \Big/ C_{internal}(t) \right) \Big/ c_{absolute}(t). \tag{11}$$

The absolute calibration coefficient $c_{absolute}(t)$ may be constant in time $c_{absolute}(t) = c_{absolute}$. A laser at the CL31 operating wavelength ($\sim 905$ nm) is sensitive to absorption of water vapour in the atmosphere which can have implications for the absolute calibration (Markowicz et al., 2008; Wiegner and Gasteiger, 2015). As the evaluation of absolute calibration techniques is beyond the scope this study, for simplicity the impact of this external calibration is neglected (i.e.

$c_{absolute}(t) = 1$ and hence $\beta = \tilde{P}^{OC}$ is assumed).

### 5.2.    Signal strength and noise

Given the noise is a critical component of the attenuated backscatter recorded, data with values below a certain signal-to-noise ratio ($SNR$) are unlikely to contain sufficient information about the state of the atmosphere. Where high-resolution observations are obtained, rolling spatial (along-range) and temporal averaging increases the signal contribution relative to

the noise. For every range gate $r$ and time step $t$, the smoothed signal is the average over a temporal window of fixed size 2 $w_t$ + 1 (with $w_t$ time steps) and a range window of fixed size 2 $w_r$ + 1:

$$\hat{\beta}_{smooth}(t, r) = (2w_t + 1)^{-1} \cdot (2w_r + 1)^{-1} \sum_{k=r-w_r}^{k=r+w_r} \sum_{h=t-w_t}^{h=t+w_t} \hat{\beta}(h, k). \tag{12}$$

Optimal window length depends on hardware characteristics (i.e. noise-levels), resolution settings for raw data acquisition and the application. Here window lengths combining to an averaging factor of about 1000 have been found suitable to prepare data for the detection of mixing height with a relatively larger temporal averaging window (i.e. $w_t = 50$; $w_r = 5$) as

features of the ABL structure show more variability in the vertical. Such large window sizes can significantly improve the $SNR$, i.e. signal strength compared to average background noise. Still, Sokół et al. (2014) suggest temporal averaging should be shorter in the morning transition period when boundary layer dynamics may vary extensively over time scales of 30 – 60 min. They use a 5 min averaging window on the data prior to their mixing height analysis.





To evaluate the quality of recorded attenuated backscatter it can be compared to the *noise floor*. When no high cirrus clouds are present, it is assumed the signal observed at the very highest range gates contains only noise (i.e. the atmospheric signal contribution is negligible). In this case, the noise floor $F$ can be defined as the mean $\bar{\beta}$ plus standard deviation $\sigma_\beta$ of the

background-corrected attenuated backscatter $\hat{\beta}$ (or signal $\hat{P}$, i.e. before range correction) across a certain set of gates from the top of the profile. Statistics are applied across these gates at the top of the profile and moving temporal windows (as in Eq. (12)):

$$F(t) = \bar{\beta}(t) + \sigma_\beta(t). \tag{13}$$

Here, the top 300 m of the profile ($N = 30$ at 10 m resolution) are used to determine the noise floor to ensure sufficient representation of the range-variability. Similar results would be obtained with slightly more range gates included, however,

given the discontinuity and increased noise levels around 7000 m (Sect. 3.1), it is not advisable to include more than the top 600 m in the calculation of the noise floor. The mean $\bar{\beta}$ across the top range gates is usually small and fluctuates around zero. However, if the background correction is not performed (Sect. 3.1) it can have a slight offset from zero and even a diurnal pattern for data acquired with firmware version 1.71 which performs the cosmetic shift based the dynamic zero-level (Sect. 2). Calculated from the entirely background corrected signal or attenuated backscatter (see Eq. (1-3)), the noise floor $F$

is nearly equal to the standard deviation $\sigma_\beta$ across the top range gates.

To ensure that profiles used for the calculation of $F$ do not contain any cirrus clouds, which could cause a physical signal even in the highest range gates at the top of the profile, the "relative variance" $RV(t,r)$ (or coefficient of variation) is used to mask cloud observations (Manninen et al., 2015). For each time $t$ and range $r$ (at the top of the profile), the relative

variance is the ratio of the standard deviation $\sigma_\beta(t,r)$ to the mean $\bar{\beta}(t,r)$, with statistics applied over moving windows (as in Eq. (12)), along range and time (here, $w_r = w_t = 3$ were used):

$$RV(t,r) = \left(\frac{\sigma_\beta(t,r)}{\bar{\beta}(t,r)}\right)^2 \tag{14}$$

If $RV(t,r)$ is sufficiently small, then the backscatter is interpreted as a true signal backscattered by atmospheric constituents of interest (e.g. cirrus clouds). Such backscatter values should not be incorporated into the calculation of the noise floor (Eq. (13)). Rather $F$ should be estimated for observations when $RV$ exceeds a threshold $T_1$, $\hat{\beta} = \hat{\beta}(RV > T_1)$. A threshold of

$T_1 = 1$ indicates that the variability exceeds the mean signal and can be used to mask strong backscatter from clouds. Times with a small number of gates (e.g. layer less than 100 m) available for calculation of the noise floor (i.e. when many of the top range gates are covered by cirrus) can be interpolated linearly in time. A day with cirrus in the top gates (Fig. 7) illustrates how the threshold $T_1$ can be applied to convert the $RV$ field (Fig. 7a) into a mask removing the cirrus signal from the attenuated backscatter (Fig. 7b). Based on the attenuated backscatter with clouds masked out (Fig. 7c), the noise floor $F$





is calculated over the course of the day (Fig. 7d) and the area missing due to the presence of cirrus is interpolated linearly over the time period where the attenuated backscatter has been masked out.

The *SNR* is calculated from smoothed, non range-corrected attenuated backscatter (Eq. (12)) and the noise floor *F*:

$$SNR(r,t) = \frac{\hat{\beta}_{smooth}(r,t)}{F(t)}. \tag{15}$$

Note that clean air with low aerosol content is dominated by molecular scattering which is below the sensitivity of a CL31 ceilometer (Sect. 2). Furthermore, thick liquid clouds have the ability to (almost) fully attenuate the ceilometer signal so any returns from above such a cloud layer (or even within it) correspond to noise rather than to real scattering from atmospheric particles or molecules. Hence, at certain heights the information content of the signal may be limited. To evaluate where the signal contribution is clearly distinguishable from the noise, Welch's t-test (Welch, 1947) is performed comparing the

distributions of $\hat{\beta}_{smooth}$ and F, assuming they are both normal. As $\hat{\beta}_{smooth}$ was found to deviate from normality below a range of about 500 m, Welch's t-test was only run for higher ranges. A p-value < 0.01 was chosen to accept (individually for each range gate and time step combination) that $\hat{\beta}_{smooth}$ significantly exceeds the noise floor at the respective time step. The acceptance level calculated for each *SNR* bin (Fig. 8) reveals a clear divide between observations with high information content and those with a magnitude comparable to the noise floor (or lower). For the LUMA sensors (Table 2) acceptance

levels of 50% - 90% correspond to SNR values of 0.05 − 0.20, which indicates the range of threshold values that can be selected depending on if a more relaxed or conservative filtering is desired. These low values can be explained by the fact that most observations with no significant signal contribution become or remain negative after the smoothing (Eq. (12)) has been applied.

The impact of averaging, background correction and noise filtering on observations taken by different sensors and firmware versions is illustrated for two case study days with: clear-sky conditions (Fig. 9, rows 1 and 2) and some boundary layer clouds present (Fig. 9, rows 3 and 4), respectively. The evolution of the ABL becomes apparent after the running average is applied (compare Fig. 9a, b). The increase in signal due to the moving average is evident in both old and new generation sensors running with different firmware versions. The impact of the background correction (Eq. (1)-(3)) has serious

implications for observations in the ABL given that the electronic background might obscure the transition between the boundary layer top and the clear atmosphere above (compare Fig. 9b, c, firmware 1.61) or the cosmetic shift might reduce the information content so strongly that certain regions within the boundary layer are lost (compare Fig. 9b, c, firmware 1.71). Here, the background correction can increase data availability. The impact of the background correction is small for new-generation sensors running with versions 2.xx. Applying the *SNR* filter based on the statistical threshold helps to

distinguish data with significant information content (compare Fig. 9c, d) so that quality can be assured for later applications (e.g. mixing height detection). Data quality of sensors running with new engine boards and versions 2.xx are clearly superior to older generations.





## 6. Summary

Ceilometers are valuable instruments with which to study not only clouds, but also the ABL and elevated layers of aerosols. Vaisala CL31 sensors provide good quality attenuated backscatter (as it is not absolutely calibrated, here it is referred to as 'raw signal'). However, to use these profiles the user needs to be aware of the instrument model's specific hardware and firmware. The following aspects are useful to consider in post-processing of ceilometer attenuated backscatter profiles:

- Initial, internal averaging of the sampled ceilometer signal is applied over two selected time intervals that depend on the user-defined reporting interval and the range for firmware versions < 1.72, 2.01 and 2.02. Data acquired with firmware 1.72 or 2.03 are more consistent than earlier versions because the whole profile (at all range gates) is treated equally with an internal averaging interval of 30 s.

- If the user defined reporting interval is shorter than 30 s, consecutive profiles partly overlap in time and are hence not completely independent.

- When averaging several profiles, a discontinuity is evident at around both 4940 m and 7000 m for all sensors and firmware versions. These regions of increased noise are introduced by the data storage procedure of the firmware which slightly changes its operating mode after a certain number of gates have been collected. Care should be taken when looking at gradients near these heights.

- Depending on firmware version, a 'cosmetic shift' is applied to the attenuated backscatter profiles. This shift should be reversed before using the entire profile for analysis. The cosmetic shift appears to be negligible for all versions except for version 1.71 in which a strong negative shift is applied to the observations.

- In addition, a range-dependent electronic background is inherent in the recorded signal and this alters the profiles systematically. The background (electronic noise plus potential cosmetic shift) values are mostly positive for older CL31 ceilometers (engine board CLE311) and slightly negative for newer hardware (CLE321); both switch sign between about 6000 – 7000 m.

- A climatology of night-time profiles is used to determine the background correction that is required. A comparison with dark-current measurements using a termination hood proves the nocturnal climatology accurately describes the background profile. Thus, the two can be considered equivalent and the background correction can be determined through either termination hood reference measurement (e.g. if profile observations are not available for a long time) or the climatology approach (e.g. if using historical data or the ceilometer site is difficult to access). Neither technique provides reliable information below about 2400 m so the profile is assumed to be range-invariant in this region.

- The background correction shows some temporal variation over the course of a day which is linked to the dynamic zero-level applied in the internal processing of the sensor. This effect can be accounted for by including an offset based on average observations in the top range gates.





- Both the range-dependent electronic background and the cosmetic shift applied may cause issues for studying the ABL because signal differences expected at the ABL top may be obliterated or the signal reduced too strongly for successful mixing height detection.

- Molecular scattering at the instrument wavelength is very weak, typically below the sensitivity of the instrument, so that the profile is dominated by the hardware-generated noise. Electronic background and cosmetic shift should be carefully evaluated before using noise for quality control purposes.

- For some instruments, 'ringing' is detected that superimposes a wave-type structure over the random noise. For the sensors evaluated, this was found in two generations of the engine board (CLE311 and CLE321), but only for transmitter type CLT321. Temporal rolling averages may enhance this ringing effect at short time scales of hours to weeks. When averaging on longer time scales of several months, the wave-structure may be removed via the averaging. This is because range gates of values with positive and negative bias are not entirely constant over time.

- Vaisala instruments have an optional setting ('*Message profile noise_h2 off*') that restricts the range correction to the signal in the lower part of the profile up to a set, critical range. It is implicitly assumed that most data at ranges beyond this critical height contain only noise. If no clouds are present in the profile, then the signal is simply multiplied by a constant, height-invariant scale factor. Where clouds are detected, the signal is actually range-corrected as usual, but only for range gates where cloud is determined to exist. To create a fully range-corrected signal for the whole vertical profile (as if '*Message profile noise_h2 on*') in the absence of clouds, the scale factor needs to be reversed and range correction applied to the observations above the critical height.

- Several artefacts may be found in the lowest range gates close to the instrument. The co-axial beam design of the CL31 ceilometer allows a complete overlap to be reached at 70 m. Below this range, an overlap correction is applied internally by the sensor. Vaisala applies another correction to observations from the first few range gates to avoid exceptionally high readings when the ceilometer's view is obstructed (e.g. by a leave on the window), At times, this *obstruction correction* introduces extremely small values at ranges < 50 m that appear unrealistically offset from the observations above this height. For some sensors, backscatter values are also slightly offset in the range of 50-80 m which can be explained by a hardware-related perturbation. The artefacts related to obstruction correction and hardware perturbation are accounted for in versions 1.72 and 2.03, but data from earlier versions need to be corrected during post-processing if observations from the near range are to be analysed. Based on climatological statistics of well-mixed atmospheric profiles, a correction procedure has been developed.

- Although CL31 output is labelled as attenuated, range-corrected backscatter, the absolute calibration might not be accurate enough for use in meteorological research. The stratocumulus or liquid cloud calibration (O'Connor et al., 2004) can be used to determine the instrument-specific lidar constant based on external properties. This allows absolute calibration to be performed during post-processing. Note that absolute calibration is included for completeness, but is not addressed here.



- As the noise in the background corrected signal is independent of range, it can be determined using the top-most range gates in the profile where the contribution of real atmospheric scattering is negligible in the absence of high cirrus clouds. The *noise floor* is here taken as the mean plus standard deviation calculated across moving windows in range and time over those top range gates. Regions containing significant aerosol or cloud should be excluded. They are efficiently masked based on their relative variance.

- To increase the signal strength, a moving average is calculated for the non range-corrected backscatter across set windows in range and time. The relation of these smoothed statistics and the noise floor defines the signal-to-noise ratio (*SNR*). The latter may be used to mask observations where the noise exceeds the actual information content of atmospheric signal. A suitable *SNR* threshold to distinguish the signal from the noise region is estimated based on Welch's t-test.

- Data quality and *SNR* of sensors running with engine boards CLE321 and firmware versions 2.xx are clearly superior to those of the old generation (CLE311).

- In response to results presented here and discussions within the TOPROF community, Vaisala released two recent firmware versions: 1.72 for sensors running with older generation hardware (engine board CLE311 and receiver CLR311) and 2.03 for sensors running with newer generation hardware (engine board CLE321 and receiver CLR321). Data collected with these two firmware versions are more consistent and show great improvement in the attenuated backscatter profiles when compared to the data from older versions.

Assuming that the sensors evaluated in this study are representative of CL31 ceilometers in general, the following conclusions can be drawn:

  i. It is advised to operate CL31 sensors with engine board CLE321 + receiver CLR321 and firmware version 2.03. Then, no corrections for cosmetic shift, electronic background or obstruction-correction artefacts are required.

  ii. Historic data collected with recent sensors (CLE321 + CLR321) and firmware versions 2.01 or 2.02 require no corrections for cosmetic shift or electronic background. Low-level artefacts should be corrected if information in the very near range (< 100 m) is of interest.

  iii. If only older hardware (CLE311 + CLR311) is available, firmware version 1.72 should be used. Then no corrections for cosmetic shift or low-level artefacts are required. Correction of the range-dependant electronic background might improve the contrast between the ABL and the clearer air above.

  iv. Historic data from CLE311 + CLR311 sensors and firmware version 1.71, require correction of the cosmetic shift plus electronic background (based on termination hood measurements or nocturnal noise-climatology). Low-level artefacts should be corrected if information in the very near range (< 100 m) is of interest.

  v. Historic data collected with CLE311 + CLR311 sensors and firmware versions 1.56 or 1.61 require no correction for cosmetic shift. Low-level artefacts should be corrected if information in the very near range (< 100 m) is of interest.



Correction of the range-dependant electronic background might improve the contrast between the ABL and the clearer air above.

Taking into account these instrument-specific aspects, the attenuated backscatter profiles from Vaisala CL31 can provide
valuable information to study not only clouds, but also the structure of the ABL or elevated aerosol layers. If data are collected with the recommended setup (or issues are being corrected for in the post-processing, e.g. applying the proposed methods) and sensors are carefully calibrated, then these observations may be used for NWP model verification and evaluation, and potentially even for data assimilation.

**Acknowledgements**

This study was funded by H2020 URBANFLUXES, NERC ClearfLo (H003231/1), European Cooperation in Science and Technology (COST) action 'TOPROF': ES1303, King's College London and University of Reading. For providing sites and other support we thank KCL Directorate of Estates and Facilities, ERG/LAQN and RGS/IBG; the many staff and students at University of Reading and KCL who contributed to data collection. We would like to acknowledge the useful discussions
within the TOPROF community (especially with Mariana Adam, Met Office, and Frank Wagner, DWD).

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




Table 1: Internal averaging interval applied in different CL31 firmware versions as a function of range r and reporting interval.

| Firmware version | Range [m] | Reporting interval | | | |
| --- | --- | --- | --- | --- | --- |
| | | 2 s | 3 – 4 s | 5 – 8 s | > 8 s |
| <1.72, 2.01, 2.02 | r < 600 | 2 s | 4 s | 8 s | 16 s |
| | 600 ≤ r < 1200 | 4 s | 4 s | 8 s | 16 s |
| | 1200 ≤ r < 1800 | 8 s | 8 s | 8 s | 16 s |
| | 1800 ≤ r < 2400 | 16 s | 16 s | 16 s | 16 s |
| | r ≥ 2400 | 30 s | 30 s | 30 s | 30 s |
| 1.72, 2.03 | r > 0 | 30 s | 30 s | 30 s | 30 s |

Table 2: Vaisala CL31 ceilometer specifications of sensor hardware, firmware, noise setting and resolution assigned by the user. The term 'H2' is discussed in Sect. 3.2.

| Sensor ID | Network | Ceilometer Engine Board / Transmitter | Firmware version | H2 | Resolution (time, range) |
| --- | --- | --- | --- | --- | --- |
| A | LUMO | CLE311 / CLT311 | 1.56, 1.61, 1.71 | On | 15 s, 10 m |
| | | CLE311 / CLT321 | 1.71, 1.72 | On | 15 s, 10 m |
| B | LUMO | CLE311 / CLT311 | 1.61 | On | 15 s, 10 m |
| | | CLE311 / CLT321 | 1.61, 1.71, 1.72 | On | 15 s, 10 m |
| | LUMO | CLE321 / CLT321 | 2.01, 2.02, 2.03 | On | 15 s, 10 m |
| D | LUMO | CLE321 / CLT321 | 2.01, 2.02, 2.03 | On | 15 s, 10 m |
| W | Met Office | CLE311 / CLT311 | 1.71 | Off | 30 s, 20 m |



Table 3: Instrument specific correction function coefficients to address systematic alterations in the lowest 100 m introduced by a hardware-related perturbation for four LUMO sensors (Table 2) in 2013 when operating with firmware versions 1.61 (A & B) and 2.01 (C & D), respectively. The intercept b and slope a are given for a linear regression by range gate: $\tilde{P}^{OC}(n)/\tilde{P}^{OC}(10) = a \cdot \tilde{P}^{OC}(5)/\tilde{P}^{OC}(10) + b$ with range gate $n \in (6, 7, 8, 9)$.

| Range gate | b (intercept) | | | | a (slope) | | | |
|---|---|---|---|---|---|---|---|---|
| | A | B | C | D | A | B | C | D |
| 6 | 1.4190 | 0.3557 | 0.0210 | -0.6552 | -0.1654 | 0.5656 | 0.8651 | 1.3990 |
| 7 | 0.8661 | 0.4197 | 0.6530 | 0.4471 | 0.1648 | 0.5099 | 0.3023 | 0.4734 |
| 8 | 0.7581 | 0.5638 | 0.8765 | 0.9456 | 0.1649 | 0.3349 | 0.0613 | 0.0147 |
| 9 | 0.8367 | 0.7739 | 0.9634 | 1.0057 | 0.1111 | 0.1830 | 0.0187 | -0.0107 |





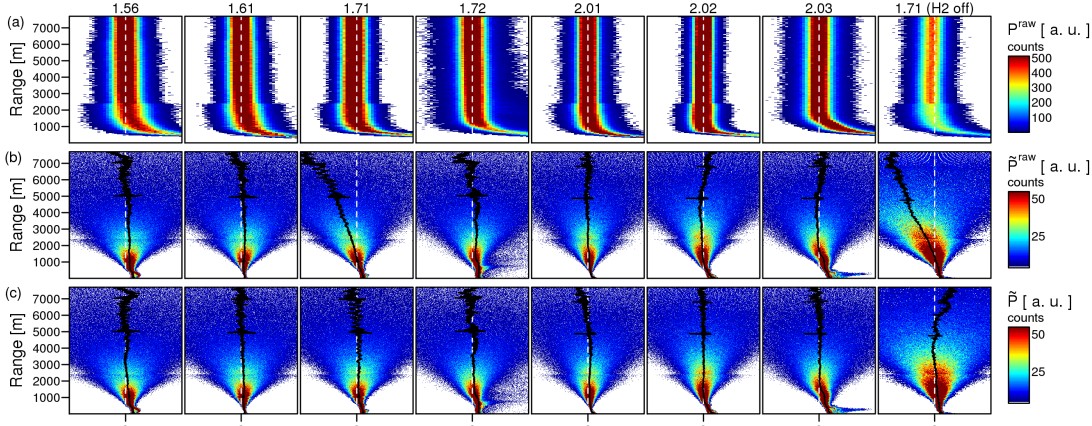

Fig. 1: Range histograms for 24 h of observations on different clear-sky days from Vaisala CL31 ceilometers operating with firmware versions (1.56 – 2.03), all with H2 setting = on (resolution: 15 s, 10 m), except the last column (firmware version 1.71, H2 setting = off, resolution: 30 s, 20 m). Rows: range histograms in arbitrary units (a.u.) of (a) not-range corrected raw signal $P^{raw}$, (b) range-corrected raw signal $\tilde{P}^{raw}$, and (c) range-corrected, entirely background-corrected signal $\tilde{P}$ (Eq. (4)). Median profiles (solid lines) are included in (b) and (c). The H2 setting described in Sect. 3.2 can be used to switch off the range correction above 2400 m for regions with no clouds present.



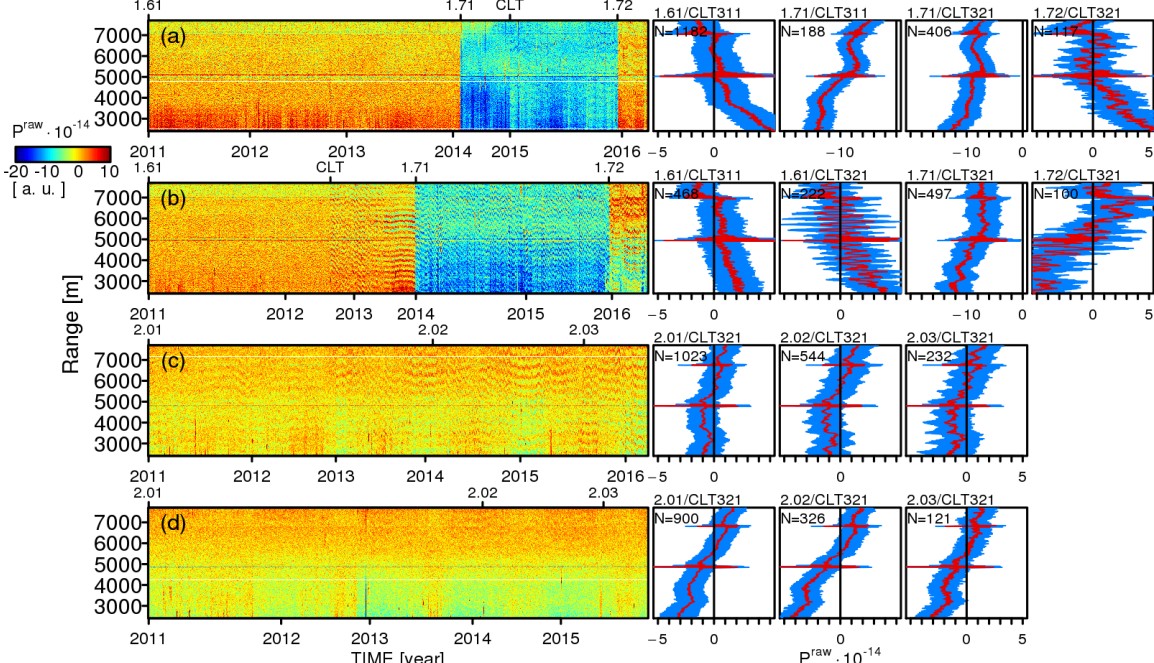

Fig. 2: Raw signal $P^{raw}$ (not range-corrected) observed with four Vaisala CL31 sensors operating with (a, b) board CLE311 (A & B) and (c, d) CLE321 (C & D), respectively (see Table 2) from January 2011 through March 2016 for range 2400 m – 7700 m. Observations (four hours around midnight, 22-02 UTC) are hourly means of profiles when: clouds detected for < 10% of the hour, no fog, average window transmission > 80%, laser pulse energy > 98% and data availability > 90%. (left) Top axis firmware updates (version 1.71, then 1.72 for sensors A & B; versions 2.02, then 2.03 for C & D) and hardware upgrades (transmitter CLT311 replaced by CLT321 for sensors A and B); (right) median profiles (with IQR shading) of all selected observations grouped by firmware version and transmitter generation, with N indicating the number of profiles.





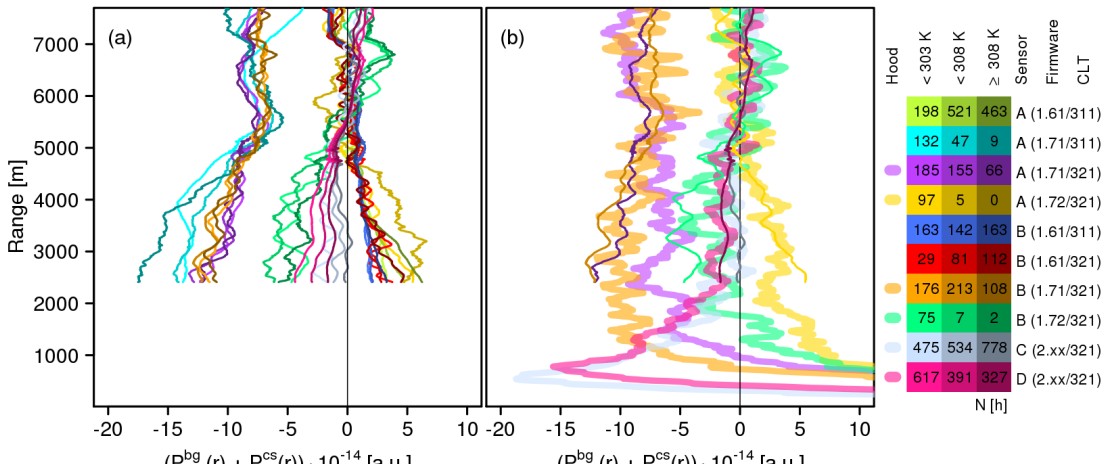

Fig. 3: Long-term median vertical profiles of range-dependent electronic background plus cosmetic shift from four Vaisala CL31 sensors (Table 2). Statistics are based on hourly mean profiles (2410 – 7700 m) of non-range-corrected raw signal $P^{raw}$ observed around midnight, January 2011 to March 2016 (same data as Fig. 2). Calculated separately for firmware version, ceilometer transmitter CLT and laser heat sink temperature combinations (see legend): ceilometers A & B operated with firmware 1.61, 1.71 or 1.72 and transmitter type CLT311 or CLT321, respectively; ceilometers C & D operated with CLT321 and firmware 2.01, 2.02, and 2.03, all are combined (2.xx) due to their similarity; laser heat sink temperature (as reported by the ceilometer) is used to subdivide profiles into three classes ($T_{laser} < 303$, $303 \leq T_{laser} < 308$ K, and $T_{laser} \geq 308$ K). Number of hourly mean profiles N [h] available for each combination of sensor, firmware, transmitter type CLT and laser temperature is listed in the legend. Profiles are smoothed vertically with a moving average over a window of (A, C, D) 210 m and (B) 310 m, given the wave-type bias did not average out sufficiently for sensor B. (a) All available climatological profiles; (b) Selected climatological profiles (solid lines) and their respective background profiles as determined by a 30 min termination hood measurement at the same setting and laser heat sink temperature class (thick lines): sensors operating with CLT321 and firmware 1.71 and 1.72 (sensors A & B) and firmware 2.03 (sensors C & D).





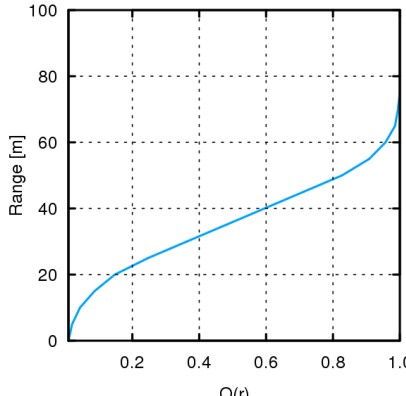

Fig. 4: Manufacturer deduced overlap function of Vaisala CL31 ceilometers using firmware versions 1.71, 1.72, 2.02, or 2.03 (older versions used an overlap function with 5 % to 10 % lower overlap values) that is applied in the lowest range gates above the instrument, derived from laboratory measurements and field observations under homogeneous atmospheric conditions. During the production process, the applicability of the function is verified for each unit. Due to the stable instrument conditions (e.g. low internal temperature variations), Vaisala expects no systematic variations of the overlap function. The error is stated to be below 10%.





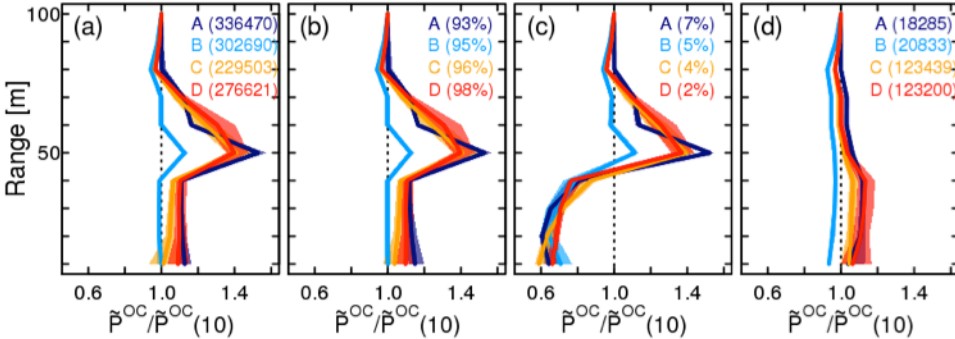

Fig. 5: Median overlap-corrected and range-corrected signal $\tilde{P}^{OC}$ of the lowest 10 range gates ($10 - 100$ m) normalised by the value at the $10^{\text{th}}$ range gate for four LUMO sensors (Table 2) with firmware versions (a-c) 1.61 (A, B) or 2.01 (C, D) in 2013 and (d) 1.72 (A, B) or 2.03 (C, D) in 2015, respectively. Statistics calculated for all profiles observed between 11-16 UTC with $\tilde{P}^{OC} < 200 \times 10^{-8}$ a.u. in the lowest 400 m: median (solid line) and inter-quartile rage (shading). Panels (b) and (c) separate the profiles from panel (a): into (b) profiles with the ratio at the $2^{\text{nd}}$ range gate, i.e. $|\tilde{P}^{OC}(2)/\tilde{P}^{OC}(10)|$, exceeding or equal to 0.8, while (c) shows the profiles with the same ratio less than 0.8. For (a, d) the total number of 15 s profiles selected is indicated by sensor A, B, C or D and in (b, c) the percentages of the values from the total number of profiles in panel (a) are given. All profiles analysed in 2015 (d) fulfil the criteria $|\tilde{P}^{OC}(2)/\tilde{P}^{OC}(10)| \geq 0.8$.





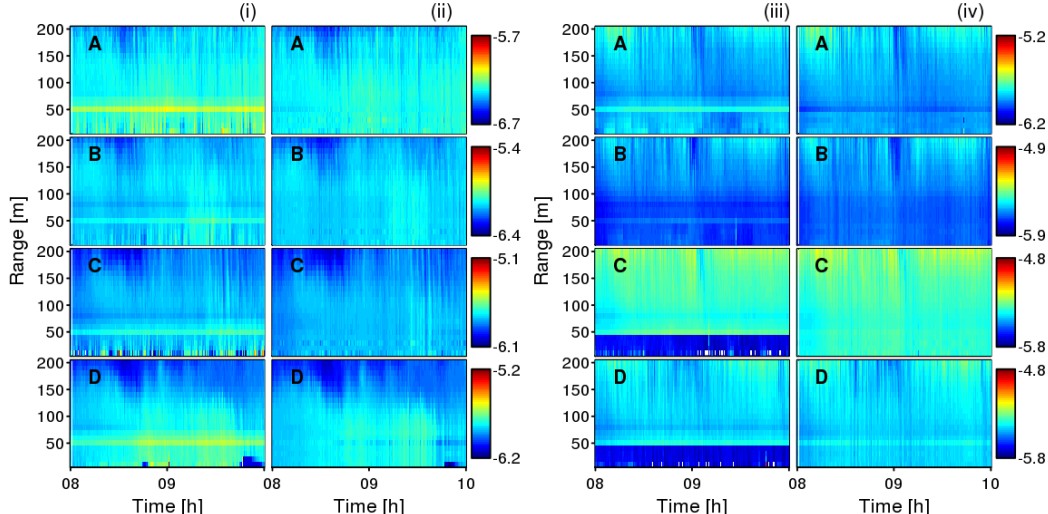

Fig. 6: Observations from four LUMO sensors (Table 2) over the first 200 m taken between 08-10 UTC on (ii, ii) 10 January 2014 and (iii, iv) 15 January 2014: (i, iii) logarithm of the entirely range-corrected and overlap corrected signal $\tilde{P}^{OC}$ [a.u.] (ii, iv) as in (i, iii), but after application of correction for low-level artefacts associated with the obstruction correction and a hardware-related perturbation (see Fig. 5 and Table 3).




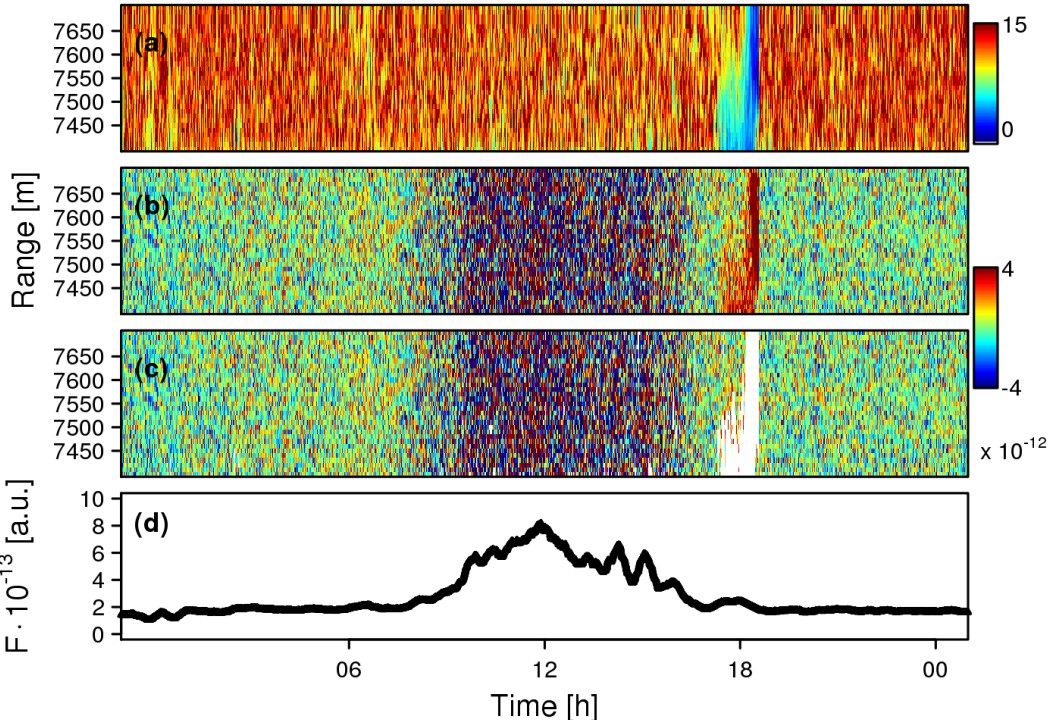

Fig. 7: Top range gates observations from ceilometer D (Table 2) with cirrus during the early evening on 1 February 2013: (a) relative variance $RV$ (Eq. (14)), (b) entirely background-corrected signal, (c) same as (b), but only including observations with $RV > 1$, and (d) time series of the noise floor $F$ (Eq. (13)) based on the cleaned signal shown in (c) with missing values interpolated linearly.





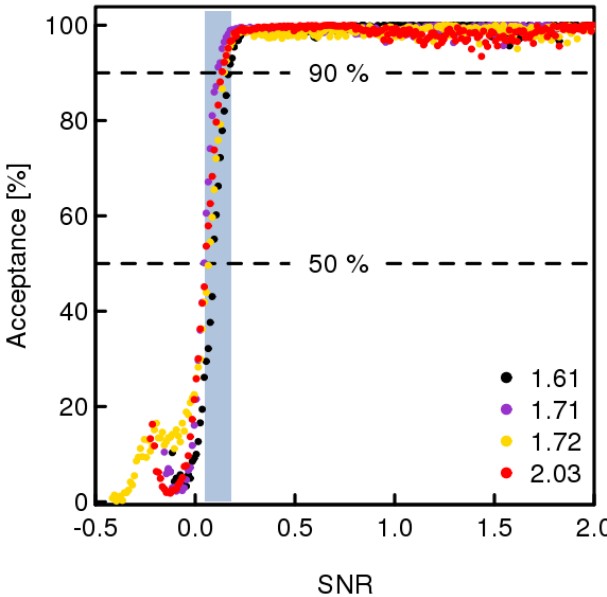

Fig. 8: Acceptance [%] based on Welch's t-test with a p-value of 0.01 of smoothed, not range corrected attenuated
backscatter (Eq. (12)) to be significantly higher than the noise floor (Eq. (14)), binned by the corresponding
signal to noise ratio (*SNR*, Eq. 14) for four selected cases (24 h observations; range 50 – 300 m shown for
simplicity) of observations taken with different firmware versions (see legend). The shaded area marks the
*SNR* region corresponding to acceptance levels of 50 – 90%.





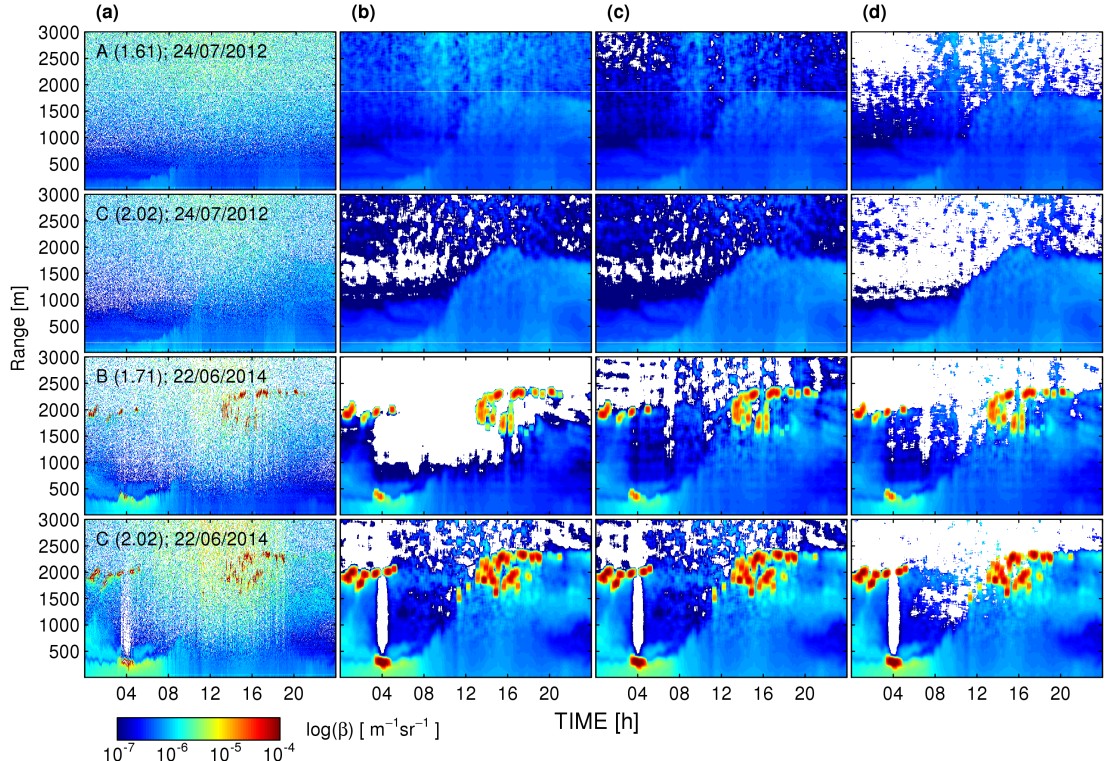

Fig. 9: Logarithm of range corrected attenuated backscatter β for the lowest 3000 m for a clear-sky day (24 July 2012; row 1 & 2) and a day with boundary layer clouds (22 June 2014; row 3 & 4) from three CL31 ceilometers (A with firmware 1.61, B with firmware 1.71, and C with version 2.02, see Table 2): (a) attenuated backscatter at recording interval of 15 s without correction for cosmetic shift and electronic background (see Sect. 3.1), (b) as in (a) with running average (101 time steps, 11 range gates) applied (Eq. (12)), (c) as in (b) but for attenuated backscatter from entirely background-corrected signal, and (d) as in (c) but filtered for $SNR > T_2$, with $T_2 = 0.18$ (see discussion on Fig. 8). Note: for simplicity the absolute calibration constant is here assumed to be $c_{absolute} = 1$ (Sect. 5.1) for all sensors. This is not necessarily expected to be a correct assumption in reality but applied to show the impact of corrections on the final product, i.e. the attenuated backscatter.