# Peer review of "Recommendations for processing atmospheric attenuated backscatter profiles from Vaisala CL31 Ceilometers"

_Atmospheric Measurement Techniques, 2016_

## Referee Comment (RC1) · Anonymous Referee #1 · 7 Apr 2016

This manuscript is a very valuable contribution which helps the scientific community to better interpret CL31 ceilometer data. This interpretation has sometimes turned out not to be trivial. Therefore, the analysis and hints given here are urgently needed.

I wonder whether there could be achieved a somehow better balance between the Introduction and the Conclusions. Otherwise, I recommend to publish this manuscript after minor revisions.

The Introduction is rather short and does not really reflect the widespread use of this type of ceilometer in scientific research. Only a few references are given. There is, e.g., a study where CL31 ceilometers are compared to older ceilometers and to temperature profiles obtained by a RASS (Emeis et al., 2009) which could be mentioned as well.

One reason why backscatter intensity from ceilometers have to be free of artefacts is that these instruments are frequently used to track the depth of the atmospheric boundary layer. The necessity could be illustrated by making reference to a paper which summarizes such tracking schemes (Emeis et al. 2008).

The Conclusions are rather long and therefore not really handy for the reader. Maybe, there could be made a split between more technical issues (which hardware to be used with which firmware) and some general conclusions being independent of specific hardware and firmware versions.

References

Emeis, S., K. Schäfer, C. Münkel, 2008: Surface-based remote sensing of the mixing-layer height – a review. Meteorol. Z., 17, 621-630. DOI: 10.1127/0941-2948/2008/0312

Emeis, S., K. Schäfer, C. Münkel, 2009: Observation of the structure of the urban boundary layer with different ceilometers and validation by RASS data. Meteorol. Z., 18, 149-154. DOI: 10.1127/0941-2948/2009/0365

---

## Referee Comment (RC2) · Anonymous Referee #2 · 15 Apr 2016

This manuscript is an excellent example of how the scientific community (COST-TOPROF) can positively influence industrial developments (Vaisala).

The given final recommendations, along with the detailed corrections procedures, can serve the CL31 ceilometer users well. Simultaneously, it is shown how a nontrivial and especially technical approach should to be undertaken, to properly use available off-the-shelf instruments.

I must agree with the Referee#1 that the weak point of the manuscript is the somewhat confusing introduction. I feel the introduction comes across as rather general, referring to scientific investigations conducted with any type of ceilometer. From one aspect, this is an interesting approach, as there are not that many technical or scientific papers on

ceilometer remote sensing. However, the authors mention to great extent, only those publications that are referring to the CL31 ceilometer. It would be more beneficial to guide the reader toward studies also performed with other ceilometer types, such as

Heese et al., Ceilometer lidar comparison: backscatter coefficient retrieval and signal-to-noise ratio determination, Atmos. Meas. Tech., 3, 1763–1770, 2010

Stachlewska et al., Ceilometer Observations of the Boundary Layer over Warsaw, Poland, Acta Geophysica, Vol. 60, No. 5, 1386-1412, 2012.

There are also comparative studies that were conducted with various types of ceilometers and/or other instrumentation or model outputs, that in my opinion, should be mentioned; to name just a few

Madonna et al., Ceilometer aerosol profiling versus Raman lidar in the frame of the INTERACT campaign of ACTRIS, Atmospheric Measurement Techniques, 8(5):2207-2223, DOI: 10.5194/amt-8-2207-2015

Emeis et al. Observation of the structure of the urban boundary layer with different ceilometers and validation by RASS data, Meteorologische Zeitschrift, Vol.18, No. 2, 149-154, 2009

Selvaratnam et al.: Comparison of planetary boundary layer heights from Jenoptik ceilometers and the Unified Model Forecasting Research Technical Report No: 605 October 13, 2015

I would however like to mention that I do appreciate how clearly and succinctly this paper is written, in particular the summary.

I hereby recommend publishing the manuscript after minor revisions.

---

## Referee Comment (RC3) · Anonymous Referee #3 · 23 Apr 2016

This manuscript presents a set of corrections to be applied to CL31 ceilometer data. It is worthy to highlight how the corrections are presented according to different firmware versions and sensors. The excellent scientific significance is evidenced by its contribution to the climatological studies which have to rely on old databases and, as it was already commented by the Anonymous Referee 2, by its contribution to the industrial developments. Despite I recommend its publication, the authors should consider the following comments:
- I agree with the previous Referees regarding the introduction and conclusion. Maybe the summary can be split into Summary with a 'list of corrections' and finally the Conclusions.

[Figure]

- Following the previous comment, the Summary surprisingly provides more information about the ringing effect than its proper section. For example, the ringing period is not provided during the discussion but it is included in the summary: Page 17 Line 9-11: 'ringing effect at short time scales of hours to weeks'. I suggest a careful revision.

- The lack of references to other papers of other type of ceilometer have been evidenced by Anonymous Referee 2. The introduction may show other corrections as the overlap correction method proposed Hervo et al., 2016 (10.5194/amt-2016-30).

- The Section 3 and Section 4 are named 'Profile corrections' and 'low-level corrections', respectively. This is confusing since the low-level correction are also profiles corrections. I suggest to change name of Sections 3 and 4 by 'Far-field corrections' and 'Near-field corrections', respectively, or similar. Additionally, the phrase 'low-level correction' is confusing (is it about near range or 'not important' corrections?). Near- and far-field corrections or near- and far-range correction may help to avoid misunderstandings.

- Page 8 Line 11: state -> states.

- Page 8 Lines 18-22: In this section it is commented that the background noise cannot be analyzed below 2400 meters because of the aerosol and humidity and because of the termination hood is unusable. Then, the background is extrapolated from the upper region. Thus, this assumption is performed in the most important region (below 2400 m) where the aerosol used to be present. Authors should include a discussion about the uncertainty due to this assumption and/or how these uncertainties may be estimated. For example, measurements performed at high stations (at mountains) may avoid the aerosol and humidity and thus, the methodology might be applied without assumptions.

- Obstruction correction: as far as I understood, this section presents a method to correct the first 10 bins of the profile. This method uses a linear fit for the last 5 bins (5th-10th) whereas the 1st, 2nd and 4th are normalized to the 3rd one. The normalization of the first four bins is based on the following argument: 'Generally, the first four gates have height-invariant factors'. I think that this phrase means that the backscatter signal

is usually height-independent in this region. Despite I agree with the sentence, 'generaly' doesn't mean 'always'. I would like to warn that the continuous and widespread application of this correction may mask real signal changes in the first meters of the atmosphere as the first convection cells (after sunrise) or different hygroscopic growth at different altitudes during fog formation. Further studies in this way should be performed to assure that no real changes are being masked.

As it said before, I strongly recommend publishing the manuscript after minor revisions.
* * *

---

## Referee Comment (RC4) · Anonymous Referee #4 · 4 May 2016

GENERAL COMMENTS

It is very good to see such a detailed description of the system, processing steps, and correction methodology. This is very helpful to the user community.

It might justify your method to look for references of how this kind of problem/filtering has been addressed generally in sensing in the past, in signal processing journals. A simple diagram and/or equation of electronics impulse response might be helpful to audiences who don't understand why this instrument noise is here. I think I've seen something like this in a reference you already mentioned, Vande Hey, 2014.

TYPOS

[Figure]

P2 L18: ALC? Categorized?

P2 L19: perform sophisticated cloud height algorithms-how are they sophisticated?

P4 L2: "solar insolation", Is this phrase redundant? Don't you mean something like either "solar radiation" or "background light increase due to insolation"?

P5 L29: is word "known" needed?

P10 L21-22: "...in the near range when the window is obstructed and a hardware-related perturbation." I think this is an incomplete sentence.

P11 L14: "their impact", perhaps "the impact of these artefacts"

CORRECTIONS

P5 L11: might illustrate range-dependent noise

P5 L12: nomenclature for Praw and Praw too similar? Nomenclature in whole section is confusing

P5 L18: clarify "amplifies difference between clouds and aerosols"

P5 L29: is "known" the correct term for the air molecular density profile?

P5 L30: "very little" should be more specific in this context, give an example of how much the molecular scattering signal would vary with density from a 1064nm research lidar (if it would be detectable there, must be some reference out there) to put this into context

P5 L29-P6 L3: Sentences starting "Molecular scattering..." going to ..."hardware generated noise" somewhat unclear and redundant. Please be more specific here-it might be sensible to address this point about molecular backscatter to someone used to working with higher specification lidars, explaining clearly the lack of sensitivity. Also there is a conference paper from Potenza that shows molecular profile can be retrieved from CHM-15k when averaging for hours at night-Binietoglou et al, can't remember

year.

P6 L35-P7 L1: "Hence if sufficient profiles are averaged, the ringing becomes less apparent in the climatology" This is counter intuitive-I would expect the ringing to become more pronounced with averaging. Can you offer any brief explanation of why this is? Is it because the fundamental ringing frequency(ies) change(s) with changes in gain, background noise, laser power, temperature?

P7 L26: "Dark Current" is an inappropriate term for this measurement, as that term typically relates only to the detector. Something like "instrument noise" should be used, as this encompasses any electronic and optical noise inside the instrument.

P7 L27 & L28: "Background profiles" and "Background Noise profiles" might also be misleading term that could be confused with background light. Try to either change this or qualify it. Not easy to do, but just make sure you clearly define your terms and are fully consistent throughout the text. Perhaps "instrument artefact signatures" or profiles? Background may be confusing, but if you use that term define it very carefully. Later P9L15 you say "electronic background". This is perhaps better. Only problem is it can also have an internal optical component, but maybe you can just mention that the first time you say this.

P10 L26: For the overlap multiplication factor, do you mean $1/(x{\to}0)$ to $1/1$ ? I guess this is pretty clear from equation 7.

P11-12: Section 4.2 and Figure 5 should be explained more clearly. Perhaps a schematic of a signal across the lowest range bins for each example of your corrections for the different criteria would help, before you show the actual signals. Just keep in mind that your audience could confuse the different effects of overlap, obstruction correction, etc., that you are trying to deal with here.

P13 L23: averaging factor of 1000, can you relate this to averaging times for typical capture settings?

P14 L4: is there a reference you can cite on how this noise floor is established? Given what you said earlier about low frequency artefacts, this could be slightly problematic. Just justify this choice given what you said earlier.

P14 L22: just to clarify, the relative variance test works because you get a small non-zero mean backscatter, which brings the ratio down, yes? This could be slightly confusing, because if we had strong signal from a thin supercooled water layer, we might find a larger ratio rather than a smaller one. But presumably this is eliminated by averaging over the extended range you've selected, is that right?

——————————————————

---

## Author Comment (AC1) · 16 Jun 2016

Response to Anonymous Referee #1

We would like to thank the referee for their positive feedback and valuable suggestions to improve the manuscript. Please see comments below.

Please note that an additional dataset is included in the revised version as these observations provide additional insights on sensor-specific characteristics (e.g. the instrument-related background signal). Further, the near-range correction (Sect. 3.4) is updated in the revised version to make it more generally applicable, i.e. the new version can also handle more complex conditions. Conclusions and recommendations

[Figure]

are updated to reflect results presented in the revised version.
* * *
This manuscript is a very valuable contribution which helps the scientific community to better interpret CL31 ceilometer data. This interpretation has sometimes turned out not to be trivial. Therefore, the analysis and hints given here are urgently needed. I wonder whether there could be achieved a somehow better balance between the Introduction and the Conclusions. Otherwise, I recommend to publish this manuscript after minor revisions.

→ We have re-written the Introduction and Summary section.

The Introduction is rather short and does not really reflect the widespread use of this type of ceilometer in scientific research. Only a few references are given. There is, e.g., a study where CL31 ceilometers are compared to older ceilometers and to temperature profiles obtained by a RASS (Emeis et al., 2009) which could be mentioned as well.

→ The Introduction still focuses on the CL31 ceilometer, however, now includes references to the Emeis et al. (2009) study which compares two CL31 sensors to each other and also to an LD-40. Further we include Haeffelin et al. (2011) who compare a CL31 attenuated backscatter profile to that of a Jenoptik CHM15K and Madonna et al. (2015) who compare a Vaisala CT25K to Jenoptik and Campbell systems.

One reason why backscatter intensity from ceilometers have to be free of artefacts is that these instruments are frequently used to track the depth of the atmospheric boundary layer. The necessity could be illustrated by making reference to a paper which summarizes such tracking schemes (Emeis et al. 2008).

→ The aspect of mixing height detection and implications of noise in the profile of attenuated backscatter is now included in the Introduction.

The Conclusions are rather long and therefore not really handy for the reader. Maybe, there could be made a split between more technical issues (which hardware to be

used with which firmware) and some general conclusions being independent of specific hardware and firmware versions.

→ We have split the Summary into sections: 'Instrument-specific characteristics and issues', 'Proposed corrections', and 'Concluding recommendations'.

References

Emeis, S., K. Schäfer, C. Münkel, 2008: Surface-based remote sensing of the mixing-layer height – a review. Meteorol. Z., 17, 621-630. DOI: 10.1127/0941-2948/2008/0312

→ Reference included: page 4, line 12

Emeis, S., K. Schäfer, C. Münkel, 2009: Observation of the structure of the urban boundary layer with different ceilometers and validation by RASS data. Meteorol. Z., 18, 149-154. DOI: 10.1127/0941-2948/2009/0365

→ Reference included: page 3, line 22

Please also note the supplement to this comment:
http://www.atmos-meas-tech-discuss.net/amt-2016-87/amt-2016-87-AC1-supplement.pdf

---

## Author Comment (AC2) · 16 Jun 2016

Response to Anonymous Referee #2

We would like to thank the referee for their positive feedback and the valuable suggestions to improve the manuscript. Please see comments below.

Please note that an additional dataset is included in the revised version as these observations provide additional insights on sensor-specific characteristics (e.g. the instrument-related background signal). Further, the near-range correction (Sect. 3.4) is updated in the revised version to make it more generally applicable, i.e. the new version can also handle more complex conditions. Conclusions and recommendations

are updated to reflect results presented in the revised version.
* * *
This manuscript is an excellent example of how the scientific community (COST-TOPROF) can positively influence industrial developments (Vaisala).

The given final recommendations, along with the detailed corrections procedures, can serve the CL31 ceilometer users well. Simultaneously, it is shown how a nontrivial and especially technical approach should to be undertaken, to properly use available off-the-shelf instruments.

→ Thank you for this comment. We have included in the summary that the initial product of the CL31 ceilometer, i.e. the cloud base height, might be readily useful without deeper understanding of the instrument-specifics, however, that special care needs to be taken when working with the attenuated backscatter.

I must agree with the Referee#1 that the weak point of the manuscript is the somewhat confusing introduction. I feel the introduction comes across as rather general, referring to scientific investigations conducted with any type of ceilometer. From one aspect, this is an interesting approach, as there are not that many technical or scientific papers on ceilometer remote sensing. However, the authors mention to great extent, only those publications that are referring to the CL31 ceilometer. It would be more beneficial to guide the reader toward studies also performed with other ceilometer types, such as

→ Given the current work focuses on Vaisala CL31 sensors, the introduction did not cover other sensor types. The Introduction still focuses on the CL31 ceilometer, however, now includes references to the Emeis et al. (2009) study which compares two CL31 sensors to each other and also to an LD-40. Further we include Haeffelin et al. (2011) who compare a CL31 attenuated backscatter profile to that of a Jenoptik CHM15K and Madonna et al. (2015) who compare a Vaisala CT25K to Jenoptik and Campbell systems.

Heese et al., Ceilometer lidar comparison: backscatter coefficient retrieval and signal-to-noise ratio determination, Atmos. Meas. Tech., 3, 1763–1770, 2010

→ Now referenced, page 2, line 27

Stachlewska et al., Ceilometer Observations of the Boundary Layer over Warsaw, Poland, Acta Geophysica, Vol. 60, No. 5, 1386-1412, 2012.

→ Now referenced, page 2, line 20

There are also comparative studies that were conducted with various types of ceilometers and/or other instrumentation or model outputs, that in my opinion, should be mentioned; to name just a few

Madonna et al., Ceilometer aerosol profiling versus Raman lidar in the frame of the INTERACT campaign of ACTRIS, Atmospheric Measurement Techniques, 8(5):2207-2223, DOI: 10.5194/amt-8-2207-2015

→ Now referenced, page 4, line 22

Emeis et al. Observation of the structure of the urban boundary layer with different ceilometers and validation by RASS data, Meteorologische Zeitschrift, Vol.18, No. 2, 149-154, 2009

→ Now referenced, page 3, line 22

Selvaratnam et al.: Comparison of planetary boundary layer heights from Jenoptik ceilometers and the Unified Model Forecasting Research Technical Report No: 605 October 13, 2015

→ Now referenced, page 2, line 20

I would however like to mention that I do appreciate how clearly and succinctly this paper is written, in particular the summary.

→ We have updated and restructured the summary to make it more accessible to the

reader.

I hereby recommend publishing the manuscript after minor revisions.

 

Please also note the supplement to this comment:
http://www.atmos-meas-tech-discuss.net/amt-2016-87/amt-2016-87-AC2-
supplement.pdf

---

## Author Comment (AC3) · 16 Jun 2016

Response to Anonymous Referee #3

We would like to thank the referee for their positive feedback and the valuable suggestions to improve the manuscript. Please see comments below.

Please note that an additional dataset is included in the revised version as these observations provide additional insights on sensor-specific characteristics (e.g. the instrument-related background signal). Further, the near-range correction (Sect. 3.4) is updated in the revised version to make it more generally applicable, i.e. the new version can also handle more complex conditions. Conclusions and recommendations

are updated to reflect results presented in the revised version.

––––––––––––––––––––––––––––––––––––––––––––––––––––––––––––––––––––––

This manuscript presents a set of corrections to be applied to CL31 ceilometer data. It is worthy to highlight how the corrections are presented according to different firmware versions and sensors. The excellent scientific significance is evidenced by its contribution to the climatological studies which have to rely on old databases and, as it was already commented by the Anonymous Referee 2, by its contribution to the industrial developments. Despite I recommend its publication, the authors should consider the following comments:

I agree with the previous Referees regarding the introduction and conclusion. Maybe the summary can be split into Summary with a 'list of corrections' and finally the Conclusions.

→ We have split the Summary into sections: 'Instrument-specific characteristics and issues', 'Proposed corrections', and 'Concluding recommendations'.

Following the previous comment, the Summary surprisingly provides more information about the ringing effect than its proper section. For example, the ringing period is not provided during the discussion but it is included in the summary: Page 17 Line 9- 11: 'ringing effect at short time scales of hours to weeks'. I suggest a careful revision. → The Summary section as been revised. Comments regarding the 'ringing' effect have been adapted.

The lack of references to other papers of other type of ceilometer have been evidenced by Anonymous Referee 2. The introduction may show other corrections as the overlap correction method proposed Hervo et al., 2016 (10.5194/amt-2016-30).

→ The Introduction has been extended and restructured, now also including the Hervo et al. reference. (page 3, line 32). It is further mentioned in the section on overlap corrections (page 14, line 21).

The Section 3 and Section 4 are named 'Profile corrections' and 'low-level corrections', respectively. This is confusing since the low-level correction are also profiles corrections. I suggest to change name of Sections 3 and 4 by 'Far-field corrections' and 'Near-field corrections', respectively, or similar. Additionally, the phrase 'low-level correction' is confusing (is it about near range or 'not important' corrections?). Near- and far-field corrections or near- and far-range correction may help to avoid misunderstandings.

→ We agree that the divide was unnecessary. Now all corrections are outlined in Section 3 'Corrections'. The correction of hardware-related perturbation and obstruction correction affecting the first range gates < 100 m is now labelled 'near-range correction' as suggested.

Page 8 Line 11: state -> states.

→ Corrected

Page 8 Lines 18-22: In this section it is commented that the background noise cannot be analyzed below 2400 meters because of the aerosol and humidity and because of the termination hood is unusable. Then, the background is extrapolated from the upper region. Thus, this assumption is performed in the most important region (below 2400 m) where the aerosol used to be present. Authors should include a discussion about the uncertainty due to this assumption and/or how these uncertainties may be estimated. For example, measurements performed at high stations (at mountains) may avoid the aerosol and humidity and thus, the methodology might be applied without assumptions.

→ Given the impact of background correction is significantly reduced below 2400 m after the range correction, the assumption of a constant background correction can be justified. Now the background profiles are also displayed as range-corrected signal (Figure 3e) and zoomed into the lowest 3000 m (Figure 3e) and further discussion regarding the use of termination hood profiles is included. As uncertainties with respect

to the background correction below 2400 m remain, a comment is included stating your suggestion that the background signal may be evaluated below 2400 m at sites where ABL aerosol and humidity is low (page 11, line 15).

Obstruction correction: as far as I understood, this section presents a method to correct the first 10 bins of the profile. This method uses a linear fit for the last 5 bins (5th-10th) whereas the 1st, 2nd and 4th are normalized to the 3rd one. The normalization of the first four bins is based on the following argument: 'Generally, the first four gates have height-invariant factors'. I think that this phrase means that the backscatter signal is usually height-independent in this region. Despite I agree with the sentence, 'generaly' doesn't mean 'always'. I would like to warn that the continuous and widespread application of this correction may mask real signal changes in the first meters of the atmosphere as the first convection cells (after sunrise) or different hygroscopic growth at different altitudes during fog formation. Further studies in this way should be performed to assure that no real changes are being masked.

→ This correction is termed 'near-range correction' in the revised version (Sect. 3.4). It was updated to account for situations when actual gradients appear in the attenuated backscatter signal, such as during hygroscopic growth periods as mentioned. The correction is now simplified, i.e. no linear-relations are applied. Instead, the revised correction now only addresses those profiles with a pronounced, artificial peak value at any given range below 100 m.

As it said before, I strongly recommend publishing the manuscript after minor revisions.

Please also note the supplement to this comment:
http://www.atmos-meas-tech-discuss.net/amt-2016-87/amt-2016-87-AC3-supplement.pdf

**Supplement:**

**Recommendations for processing atmospheric attenuated backscatter profiles from Vaisala CL31 Ceilometers**

Simone Kotthaus[1], Ewan O'Connor[1,2], Christoph Münkel[3], Cristina Charlton-Perez[4], Martial Haeffelin[5], Andrew M. Gabey[1], and C. Sue B. Grimmond[1]

[1] Department of Meteorology, University of Reading, Reading, RG6 6BB, UK

[2] Finnish Meteorological Institute, Helsinki, 00101, Finland

[3] Vaisala GMBH, Hamburg, 22607, Germany

[4] Met Office, Meteorology Building, University of Reading, Reading, RG6 6BB, UK

[5] Institute Pierre Simon Laplace, Centre National de la Recherche Scientifique, Ecole Polytechnique, Palaiseau, 91128, France

*Correspondence to*: Simone Kotthaus (s.kotthaus@reading.ac.uk)

**Abstract.** Ceilometer lidars are used for cloud base height detection, to probe aerosol layers in the atmosphere (e.g. detection of elevated layers of Saharan dust or volcanic ash) and to examine boundary layer dynamics. Sensor optics and acquisition algorithms can strongly influence the observed attenuated backscatter profiles, therefore, physical interpretation of the profiles requires careful application of corrections . This study addresses the widely deployed Vaisala CL31 ceilometers. Attenuated backscatter profiles are studied to evaluate the impact of both hardware generation  and  firmware version. In response to this work and discussion within the CL31/TOPROF user community (TOPROF  European COST Action aiming to harmonise ground-based remote sensing networks across Europe), Vaisala released new firmware (versions 1.72 & 2.03) for the CL31 sensors. These firmware versions are tested against previous versions showing that several artificial features introduced by the data processing have been removed. Hence, it is recommended to use this recent firmware for analysing attenuated backscatter profiles. To allow for consistent processing of historic data, correction procedures have been developed that account for  artefacts detected in data collected with older firmware. Furthermore, a procedure is proposed to determine and account for the instrument-related background signal from electronic and optical components necessary for using attenuated backscatter observations from any CL31 ceilometer. Recommendations are made for the processing of attenuated backscatter observed with Vaisala CL31 sensors, including the estimation of noise which is not provided in the standard CL31 output. After taking these aspects into account, attenuated backscatter profiles from Vaisala CL31 ceilometers are considered capable of providing valuable

information for a range of applications including atmospheric boundary layer studies, detection of elevated aerosol layers, and model verification.

**Notation**

$\beta$ — attenuated backscatter

$P$ — signal (derived from $RCS$ by reverting range correction)

$P\,r^2$ — range-corrected signal (equal to $RCS$)

$\hat{P}$ — background corrected signal

$P^{bg}$ — background signal

$P^{bgi}$ — instrument-related background signal

$P^{cs}$ — cosmetic shift

$r$ — range

$RCS$ — range-corrected signal

**1.  Introduction**

Ceilometer lidars are widely used to characterise clouds (Illingworth et al., 2007). Sophisticated cloud base height detection is found to provide reliable estimates, with  multiple cloud layers identified (Martucci et al., 2010). Although originally developed as 'cloud base recorders' their attenuated backscatter profiles can also provide information on rainfall (Rogers et al., 1997), formation and clearance of fog (Haeffelin et al., 2010), drizzle properties (when combined with cloud radar; O'Connor et al., 2005) and for the study of aerosols, including elevated layers of Saharan dust (Knippertz and Stuut, 2014), biomass burning (Mielonen et al., 2013) or volcanic ash (e.g. Marzano et al., 2014; Nemuc et al., 2014; Wiegner et al., 2012), and particles dispersed within in the atmospheric boundary layer (Tsaknakis et al., 2011). Using aerosols as a tracer, boundary layer dynamics, including mixing height and the formation of residual layers, can be inferred from ceilometer attenuated backscatter observations (e.g. Münkel et al., 2007; Stachlewska et al., 2012; Selvaratnam et al., 2015). As they can operate automatically for long periods without maintenance or human intervention even in extreme climates (Bromwich et al., 2012), they are widely deployed operationally by national meteorological services (NMS, e.g. www.dwd.de/ceilomap) and long-term research campaigns (e.g.  micromet.reading.ac.uk).

Although ceilometers are regarded as the most basic automatic lidars (Emeis, 2010), they detect the location and extent of aerosol layers and to derive the

aerosol backscatter coefficient, provided signal-to-noise ratio (*SNR*) is sufficient and a careful calibration is applied (e.g. Jenoptik CHM15K, Heese et al., 2010; Wiegner et al., 2014). Observations from ceilometers are highly valuable for the evaluation of numerical weather prediction (NWP) and air-quality models (Emeis et al., 2011b) and are increasingly used in forecast verification. Several NMS and research centres are currently evaluating the potential of using ceilometer profile observations for data assimilation (Illingworth et al., 2015).

This wide range of applications requires careful quality control of the observed attenuated backscatter to ensure reliable data for analysis. The European COST Action TOPROF (http://www.toprof.imaa.cnr.it/) works in close collaboration with E-Profile (http://www.eumetnet.eu/e-profile) to develop protocols for quality control and quality assurance (Illingworth et al., 2015) of observations from automatic lidars and ceilometers (ALC). The E-Profile programme of the Network of European Meteorological Services (EUMETNET) aims to facilitate the exchange of observational data by harmonising the ALC networks across Europe. As ceilometers are manufactured by several companies, the sensor optics, hardware components and software algorithms may differ significantly. Discussions in the TOPROF community have revealed the importance of a detailed understanding of instrument specifics to identify the necessary processing steps enabling appropriate interpretation and harmonisation of the final data products. For example, the extensive CeiLinEx2015 inter-comparison campaign (www.ceilinex2015.de) was devised by TOPROF members to evaluate attenuated backscatter and cloud base height products from a range of ceilometer models from several manufacturers (including Lufft/Jenoptik, Campbell Sci., and Vaisala). This study addresses the commonly deployed Vaisala CL31 ceilometers. Earlier Vaisala ceilometer models include LD40 and CT25K; the CL51 is the most recent model.

Emeis et al. (2011a) report that attenuated backscatter from Vaisala CL31 ceilometers portrays structures in the atmospheric boundary layer (ABL) consistent with temperature and humidity profiles observed by radiosondes and a sodar RASS system. Initial evaluation of CL31 attenuated backscatter observations for quantitative aerosol analysis (Sundström et al., 2009) suggests accuracy might be sufficient in the ranges near the instrument if certain systematic artefacts found in the profiles can be removed or accounted for. McKendry et al. (2009) suggest that, under clear-sky conditions, the CL31 has the capability to 'detect detailed aerosol layer structure (such as fire or dust plumes) in the lower troposphere' that is consistent with the aerosol structure detected by an aerosol research lidar (CORALNet-UBC). However, comparing a Vaisala LD40 and two CL31 ceilometers, Emeis et al. (2009) show that attenuated backscatter may vary distinctly between these sensors, with clear implications for their representation of ABL structures. As these differences are manifested in vertical structures rather than as a simple offset they cannot be explained by a lack of absolute calibration. Emeis et al. (2009) state 'internally generated artefacts from the instrument's software' could play a role, however they refrained from providing further details.

While software-related artefacts might contribute to the differences, the discrepancy between the attenuated backscatter profiles observed by the two CL31 sensors tested (Emeis et al., 2009) might also be explained by the hardware-related (electronic or optical) background signal. Recent work on a Halo Doppler lidar suggests such background signal features could be corrected for during post-processing (Manninen et al., 2016).

Incomplete optical overlap can be corrected for, however uncertainties may remain. Recent research shows for example, that the overlap function of a Lufft CHM15K is slightly temperature dependent (Hervo et al., 2016). Due to the co-axial beam design, the full optical overlap for the CL31 is reached at low ranges (Münkel et al., 2009) which can be beneficial when studying meteorological processes in the lowest part of the atmosphere, such as fog, haze or aerosols emitted at the earth's surface. For example, in comparison to a LD40 which reaches complete overlap only at 200 m, the CL31 has an advantage in detecting low, stable layers (Emeis et al., 2009). ~~periods without maintenance or human intervention even in extreme climates (Bromwich et al., 2012), they are widely deployed operationally by NMS (e.g. www.dwd.de/ceilomap) and long-term research campaigns (e.g. www.met.reading.ac.uk/micromet). International initiatives, such as the E-Profile programme (http://www.eumetnet.eu/e-profile) of the Network of European Meteorological Services (EUMETNET), aim to facilitate the exchange of observational data by harmonising the ALC network across Europe. The European COST Action TOPROF (http://www.toprof.imaa.cnr.it/) works in close collaboration with E-Profile to ensure that the necessary steps of quality control are incorporated into the common data processing (Illingworth et al., 2015). As ceilometers are manufactured by several companies, the sensor optics, hardware components and software algorithms may differ significantly. Discussions in the TOPROF community have revealed the importance of a detailed understanding of instrument specifics to identify the necessary processing steps~~ that enable appropriate interpretation and harmonisation of the final data products.

Although Vaisala suggests that the attenuated backscatter profile is reliable down to the first range gate, Sokół et al. (2014) document a distinct local minimum in CL31 attenuated backscatter observations at the $4^{th}$ range gate persisting throughout their whole observational campaign. As others have found artefacts in CL31 profiles below 70 m (e.g. Martucci et al. 2010; Tsaknakis et al. 2011) these lowest ranges are often excluded during processing. Sundström et al. (2009) evaluate the applicability of CL31 observations for quantitative aerosol measurements and conclude that the artefacts in the range gates near the instrument are a major source of uncertainty. van der Kamp (2008) smooths out systematic artefacts by strong vertical averaging; however, this removes the possibility of identifying any atmospheric features close to the surface.

Various techniques have been developed to infer the mixing height from the shape of the attenuated backscatter profiles from ceilometers (Emeis et al., 2008; Haeffelin et al., 2012). While detection algorithms vary, all methods exploit the fact that aerosol concentrations (and atmospheric moisture if boundary layer clouds are absent) are typically significantly higher in the ABL compared to the free atmosphere above. This causes a distinct decrease in attenuated backscatter at the boundary layer top, provided that the signal-to-noise ratio (*SNR*) is sufficiently large up to this height.

A series of studies have successfully used CL31 observations to detect mixing height (e.g. Münkel et al., 2007; van der Kamp and McKendry, 2010; Eresmaa et al., 2012; Sokół et al., 2014; Tang et al., 2016), often reporting an increased performance under convective conditions that ensure the backscattering aerosols are well-dispersed. However, Eresmaa et al., (2012) report that fitting an idealised profile to the observed attenuated backscatter from a CL31 may be challenging where noise levels are high. As the CL31 operates with a very low-powered laser, its noise levels may be higher than that found for other ALC systems (e.g. cf. a Jenoptik CHM15K, Haeffelin et al., 2012). Madonna et al. (2015) evaluate the profiling ability of several ALC from different manufacturers (i.e. a Jenoptik CHM15K, a Vaisala CT25K and a Campbell CS135s) against a MUSA advanced Raman lidar during night-time. They conclude that the attenuated backscatter coefficient generally is in good agreement with the reference measurement for the CHM15K while the CS135s shows good agreement only for small values and the CT25K tends to underestimate, which may be related to the overall lower $SNR$ of the latter two sensors. If noise levels are too high within the ABL, as reported e.g. by Haeffelin et al. (2012) for a case study of a Vaisala CL31 ceilometer at the SIRTA site near Paris, the signal might not be sufficient to detect the top of the ABL. de Haij et al. (2006) apply an $SNR$ threshold to restrict observations from a Vaisala LD40 ceilometer to be used for mixing height detection. Such filtering based on $SNR$ diagnostics presents a useful tool to differentiate measurements containing significant atmospheric signal from observations dominated by instrument noise and atmospheric noise induced by solar radiation.

Neither the $SNR$ nor the noise inherent in each profile is provided in the output of ALC. Xie and Zhou (2005) propose a method for $SNR$ calculations for lidar observations whereby the signal profile is approximated by a linear fit to the readily averaged profile along set range bins and assigning the deviations from that fit to the noise. Markowicz et al. (2008) apply this method to observations of a Vaisala CT25K averaged over 200 s. These $SNR$ values indicate the observations are only reliable within the ABL (absence of clouds) and it is stated that an $SNR = 10$ marks 'a limiting value of detection' (Markowicz et al., 2008). Assuming there are no temporal variations in the atmosphere probed by several consecutive observations (e.g. over a few minutes), the standard deviation at each range gate could be used as a noise estimate of the respective average if high-temporal resolution measurements are recorded (Xie and Zhou, 2005). Assuming the noise is range-invariant before the range correction, a noise estimate for the whole profile could be estimated based on observations where the signal contribution is negligible, e.g. based on the topmost range gates under the absence of high-clouds and aerosol layers. Heese et al. (2010) use to the highest range gates to calculate a noise value for each profile for a Jenoptik CHM15K assuming the signal noise follows Poisson statistics as typically assumed for photon counting detectors. Vaisala sensors operate with an avalanche photo diode, so that the noise cannot be interpreted as a counting error. The $SNR$ increases significantly when high-resolution observations are averaged over certain time and/or range windows. Using a Gaussian smoothing method on observations of a Jenoptik CHM15K, Stachlewska et al. (2012) find that the $SNR$ significantly increases if the width of range windows is increased linearly. However, they remark that this may result in extensive

computing time. In addition, excessively large smoothing windows may reduce the detectability of sharp features (Haeffelin et al., 2012).

Despite the evidence that attenuated backscatter profiles are a complex data product that might have to be carefully evaluated before being used to draw conclusions on the probed atmosphere, no guidelines are available to ensure systematic quality control and quality assurance (QAQC). This study documents the important processing steps that should be considered when analysing attenuated backscatter profiles from Vaisala CL31 . Observations from three ALC networks (Sect. 2) are used to illustrate relevant data processing aspects  (Sect. 3) ). Depending on the firmware version, the CL31 instrument internal processing may introduce certain artefacts that should be accounted for if the attenuated backscatter is required for analysis. It is shown how the signal strength can be used for quality assurance (Sect. 4) and findings are summarised in the form of recommendations for the processing of CL31 profile observations (Sect. 5).

**2. Instrument description**

The Vaisala CL31  transmits a very short pulse of 110 ns (corresponding to an effective pulse length of about 16.5 m, e.g. Weitkamp, 2005). The receiver uses an avalanche photo diode (APD) detector to record the returned signal. The instrument oversamples the backscattered signal at a temporal rate which corresponds to the range resolution setting. The reported range $r$ (i.e. distance from the instrument) is range to the centre of a range gate.  3 MHz Gaussian low-pass filtering by the instrument extends and shapes the pulse response . Different vertical resolutions can be achieved depending on the sample rate. For example, a sample rate of 15 MHz is required to achieve a range resolution of 10 m, where the first observation reported at 10 m is backscattered signal for 5-15 m from the ceilometer. Every 2 s, $2^{14}$ laser pulses are emitted with a frequency of 10 kHz which takes about 1.64 s. After this period there is an idle time of 0.36 s used to perform the cloud base detection algorithm before the next set of $2^{14}$ laser pulses is emitted. After a certain number of gates have been sampled, the firmware slightly changes operation mode; thus, regions of increased noise are introduced into the backscatter profiles at two ranges: $\sim$ 4940 m and $\sim$ 7000 m. Samples collected during the 2 s intervals are averaged over certain internal intervals to create the reported signal $\tilde{P}$ at a rate defined by the reporting interval selected by the user (2 – 30 s). The internal averaging interval is specific to the firmware. (see below).

The spectral wavelength of the laser diode used in the Vaisala CL31 is $905 \pm 10$ nm at 298 K, as stated by the laser manufacturer. Vaisala finds the uncertainty of the nominal centre wavelength to be well below 10 nm. Typical spectral width (Full Width at Half Maximum, FWHM) is 4 nm. Lasers produced from the same wafer agree in terms of the centre wavelength, however, the exact centre wavelength is unknown to the user. For a specific laser the centre wavelength is slightly temperature-dependent (0.3 nm $K^{-1}$). The CL31 system heater near the laser transmitter serves to stabilise the laser

temperature in cold environments. Further, both window transmission and laser pulse energy can have an impact on the attenuated backscatter signal. The laser heat sink temperature (denoted in the CL31 output as the *laser temperature*), window transmission and laser pulse energy are therefore monitored and reported continuously. Status information (i.e. diagnostics, warnings and alarms) is included in the data message which helps to identify whether maintenance is required

5 (e.g. window needs cleaning, transmitter is failing). In addition to the detected cloud base height, the CL31 can be set to report a profile of range-corrected 'attenuated backscatter'. However, as these values lack absolute calibration (see Sect. 4.1), observations are here referred to as the 'reported range-corrected signal' (*RCS*, for details on range correction see Sect. 3.2).

10 The detector of the CL31  responds to the backscattering of the laser pulse  from molecules, aerosols, rain drops and both liquid and ice cloud particles. It also responds to noise originating from both external (e.g. daytime solar radiation) and internal (e.g. electronic) sources. The hardware-related noise is larger than the Rayleigh signal associated with clear air so that the latter is too small to be distinguished. Vaisala states that the variance of the electronic noise signal is range-independent. The background light from solar radiation increases the current through

15 the APD, but as the amplifiers are AC-coupled, the relatively slowly varying solar signal (almost DC) does not get to the A/D converter . (The AC-coupling time constant is 1 ms., i.e. the AC-coupling works as a high-pass filter with 159 Hz corner (-3 dB) frequency). This filtering results in a variable zero-bias-level (i.e. noise has negative and positive values)  that accounts for temporal variations in the atmospheric background signal. While the AC coupling removes the low frequency signal from varying solar radiation, the latter still increases signal noise (shot noise in

20 APD due to DC current). For short data acquisition intervals, backscatter values can be below zero. Electronic noise is also a function of system properties (e.g. detector temperature, transmitter lens area; Gregorio et al., 2007; Vande Hey, 2014) and can therefore be analysed by the manufacturer prior to field deployment. Heaters provide partial thermal stabilisation of the laser and detector system  in cool or cold conditions.

25 The Vaisala CL31 firmware has been modified over time along with certain developments in the hardware, i.e. the receiver (CLR) and engine board (CLE) where the internal processing takes place. These updates have resulted in the creation of a range of firmware versions. For CLE311 + CLR311, the firmware versions 1.xx are used, while sensors with CLE321 + CLR321 run firmware versions 2.xx. Changing the ceilometer transmitter CLT generation is not connected to a change in firmware. The internal averaging interval differs slightly with firmware version (Table 1). In Vaisala CL31

30 firmware versions below 1.72 and versions 2.01, and 2.02, the internal averaging interval is set to 16 s for range gates below 2400 m if the reporting interval is greater than 8 s. For reporting intervals between 5 – 8 s, the internal averaging interval is set to 8 s below 1800 m and 16 s between 1800 – 2400 m, respectively. For reporting intervals below 5 s internal averaging below 1200 m is 4 s; only for the minimum reporting interval of 2 s, internal averaging is set to 2 s below 600 m. Above 2400 m, the internal averaging is 30 s for all reporting intervals. In firmware 1.72 and 2.03, the internal averaging

interval is 30 s for the entire profile and does not change with reporting interval (Table 1). If a reporting interval is selected that is shorter than the internal averaging interval, consecutive profiles overlap in time and are hence not completely independent.

Observations from three ceilometer networks (Table 2~~) are used in this study to illustrate aspects of the data acquisition and processing of Vaisala CL31 ceilometers. The London Urban Micromet Observatory (LUMO; www.met.reading.ac.uk/micromet) is a measurement network collecting observations of many atmospheric fields to investigate climate conditions within and around Greater London, UK (for interactive map see http://www.met.reading.ac.uk/micromet/LUMA/Network.html). The Met Office operates an ALC network (http://www.metoffice.gov.uk/public/lidarnet/lcbr-network.html) across the UK with different manufacturers/models, including Vaisala CL31 ceilometers. Four sensors of the LUMO network in central London and one Met Office sensor located 60 km west of central London are used here.~~

) are used in this study to illustrate aspects of the data acquisition and processing of Vaisala CL31. The London Urban Micromet Observatory (LUMO; micromet.reading.ac.uk) is a measurement network collecting observations of many atmospheric fields to investigate climate conditions within and around Greater London, UK (for interactive map see http://micromet.reading.ac.uk/micromet/LUMA/Network.html). The Met Office operates an ALC network (http://www.metoffice.gov.uk/public/lidarnet/lcbr-network.html) across the UK with different manufacturers/models, including Vaisala CL31. A CL31 is operated by Meteo France at the SIRTA site at Palaiseau, France, for atmospheric research activities (Haeffelin et al., 2005; http://www.sirtae.fr). Four sensors from the LUMO network in central London, one Met Office sensor located 60 km west of central London, and the Meteo France/SIRTA sensor are used here.

Long-term observations are available from four CL31 ceilometers with different generations of hardware and various firmware versions. Over time, the LUMO network firmware versions have changed from the first LUMO sensor deployed in 2006 with version 1.56 (Table 2). Sensors A and B are the old hardware generation with the CLE311 board, as is the Met Office sensor W, while LUMO sensors C and D and the SIRTA sensor S have engine boards CLE321. For both sensors A and B, the transmitter has been upgraded from CLT311 to CLT321 during their operation, and for sensor S the transmitter CLT321 was replaced by a spare part of the same generation. While the LUMO sensors are set to acquire data every 15 s with a vertical resolution of 10 m, data from the Met Office ceilometer have a resolution of 30 s and 20 m, and the SIRTA ceilometer captures data every 2 s with a range resolution of 5 m. Analysis presented here uses block averages over 30 s and 15 m of the SIRTA ceilometer data.

**3.    Corrections**

**3.1.   Background correction**

~~While the amount of backscattered signal depends on the distance of the source to the instrument, the sources of noise should, in principle, be range independent. However, given that the time dependence of the data acquisition is linked to the spatial domain, some range dependence may be found in the noise, i.e. the signal *background $P^{bg}$*. As CL31 measurement design accounts for temporal variations in the background noise by introducing a variable zero level (Sect. 2), the 'raw' signal is inherently partly background corrected. Thus, only the range dependent component of the background $P^{bg}(r)$ needs to be accounted for to derive the entirely background-corrected signal $\hat{P}$. $\tilde{P}^{raw}$ denotes the range-corrected values reported by the ceilometer and $P^{raw}$ is the signal after reverting the range correction (see Sect. 3.2).~~

The backscattered signal detected by an ALC generally consists of actual signal contributions from atmospheric attenuation, the atmospheric background signal associated with scattered solar radiation and the instrument-related background signal (Cao et al., 2013). Here, 'background signal' is used to describe systematic contributions from solar radiation or instrument-components (including hardware and software). The CL31 measurement design accounts for the temporal noise bias induced by varying solar radiation by introducing a variable zero-bias-level (Sect. 2). The atmospheric background signal still contributes to the noise in the profile. On average, the range-corrected signal reported (*RCS*, labelled 'range and sensitivity normalised attenuated backscatter' in CL31 output) is inherently corrected for the impact of atmospheric background signal $P^{bga}(r)$ and only the instrument-related background signal $P^{bgi}(r)$ needs to be accounted for to derive the background-corrected signal $\hat{P}$. Given that the time-dependence of the data acquisition is linked to the spatial domain, the instrument-related background signal may vary with range while stabilisation procedures (e.g. heaters) aim to reduce its temporal variability. Here, as temporal variations due to the solar background light are removed, the remaining temporal variations are considered to represent noise both from hardware and atmospheric background signal.

The instrument-related background signal $P^{bgi}(r)$ can combine effects associated with the electronic or optical components and those associated with internal processing by the instrument. Vande Hey (2014) discuss effects related to electronic noise including impulse response for a Campbell CS135s, a system very similar to the Vaisala CL31. A specific processing procedure implemented in some CL31 firmware versions was found to alter the profile of the background  signal systematically and is here treated separately from $P^{bgi}(r)$. This processing shifts the signal artificially so that the background signal of the respective data is biased and is no longer centred on zero (e.g. data collected with version 1.71 have more negative than positive values). This is applied  to improve detection of cloud base height as it amplifies differences between the signal backscattered from cloud droplets and areas with low concentration of atmospheric scatterersalso~~facilitates visual interpretation of clouds based on the

backscattered signal. Hereafter, this bias is referred to as 'cosmetic shift' $P^{cs}$$(r)$. Thus, to derive the entirely background-corrected signal from CL31 output, the complete background  $P^{B}$ composed of range-dependent, instrument-related background signal $P^{bgi}(r)$ and the cosmetic shift $P^{cs}(r)$, needs to be accounted for:

$$\hat{P}(r) = P(r) - P^{bg}(r) = P(r) - P^{bgi}(r) - P^{cs}(r). \tag{1}$$

For data collected with firmware 1.72 or 2.03, no cosmetic shift is incorporated ($P^{cs}(r) = 0$) so that the complete background correction is represented by the range-dependent , instrument-related background signal ($P^{bg}(r) = $$P^{bgi}(r)$). The impact of  background signal and cosmetic shift on the reported signal is illustrated using observations from different clear-sky days (no elevated dust, aerosol layers or cirrus) from CL31 sensors running a range of firmware versions (Fig. 1a). Under such conditions  the only source of atmospheric signal above the  (ABL is very weak molecular and aerosol scattering.  In practice, molecular scattering at the instrument wavelength is very weak, typically below the sensitivity of the instrument (Sect. 2), so that profiles consist only of the average total background signal and the noise. As the atmospheric background signal only contributes to the noise, no systematic differences in the shape of the observed profiles would be expected and obvious departures from zero can be associated with data acquisition and processing, i.e. instrument-related background signal and potential cosmetic shift.

A suitable method to identify discrepancies in the profile shape is to create signal-range histograms (Fig. 1a) using 24 h (or more) of data. The most obvious effect revealed by the range histograms is a step-change in the width of the distributions at 2400 m evident for all firmware versions, apart from 1.72 and 2.03. This step-change is introduced by the averaging of the sampled signal that is applied internally by the instrument's firmware (Sect. 2, Table 1). The decrease in averaging time for range gates < 2400 m performed for earlier firmware increases the signal noise (see Sect. 4.2). Data acquired with version 1.72 or 2.03 are more consistent across all range gates as the whole profile is treated equally with an internal averaging interval of 30 s.

The range histograms (Fig. 1a) show the impact of the incomplete background correction, i.e. instrument-related background signal and cosmetic shift are not  accounted for. Both cause a systematic pattern in the observed profiles illustrating the range-dependence of the background signal. The cosmetic shift is particularly strong for version 1.71. To capture both background effects, profiles are analysed during

times when atmospheric variations are expected to be small and instrument conditions are stable (Fig. 2).  $P$-profiles are extracted and averaged hourly when noise induced by solar radiation is absent (4 h around midnight), when cloud cover is low (< 10% of the hour), no fog is present, the window transmission is reasonable (> 80% on average), laser pulse energy is high (> 98% of nominal energy), and sufficient data are available (> 90% of the hour; data gaps may occur due to maintenance or problems with data acquisition such as power cuts). Only range gates > 2400 m are analysed to avoid the impact of changing internal averaging intervals (Sect. 2) at this critical range (Fig. 1a) and to minimise the signal from the ABL (unlikely to extend above 2400 m over London around midnight). Median vertical profiles (with inter-quartile range (IQR) shading) are displayed for common setup conditions (Fig. 2,  right), i.e. grouped by  combinations of sensor, firmware and transmitter (CLT), respectively. Engine board and receiver were not changed for any of the sensors during their operation.

The night-time profile climatology  (Fig. 2) reveals a small temporal variability with a seasonal cycle (amplitude ~ 50%) that indicates a temperature dependence of the instrument-related background signal. Several features appear distinct in the spatial domain (Fig. 2) at certain range gates. For all sensors and firmware versions, a discontinuity is evident just below below 5000 m and at around 7000 m. These regions of increased noise are introduced by the data storage procedure (Sect. 2). Changing hardware components affects the instrument-related background signal even if the same model is swapped in. For example, exchanging the transmitter of sensor S by a part of the exact same model (CLT321 exchanged in September 2015, Fig. 2e) resulted in a clear increase of the background signal below about 4000 m. As it cannot be guaranteed that the new transmitter has the exact same characteristics as the one replaced (Sect. 2), a slight change in wavelength might explain this shift.

For sensor B (Fig. 2b), the change in transmitter from CLT311 to  CLT321 also altered the profile of the background signal, mainly by introducing a systematic pattern along the range . A wave-like structure appears superimposed over the random noise for ceilometer B when operating with transmitter CLT321 (Fig. 2b). A similar effect is detected in observations from ceilometer C in general (Fig. 2c) and to some extent in ceilometer S after the transmitter was changed (Fig. 2e). Such 'ripple' patterns are introduced by a physical effect which overlays a vertically alternating positive and negative bias on top of the signal noise. While this wave-type bias tends to be similar for successive profiles (regions with positive and negative amplitude overlap), it is not entirely constant over  the course of a day because it is slightly affected by attenuation by clouds and ABL-particles. As shown, this ripple is sensor-specific (e.g.  higher frequency  detected for sensor B than C, Fig. 2b, c). While ripple may occur for ceilometers with both  CLE311 + CLR311 (Fig. 2b) and

CLE321  CLR321 (Fig. 2c, e) engine board plus receiver combinations, only sensors operating a transmitter of type CLT321 were found to have the ripple effect (of those tested). The firmware version does not affect this wave-type bias as it is solely a hardware-related (electronic and/or optical) contribution to the background signal $P^{bgi}(r)$. At the time of publication of this paper, Vaisala could not fully explain the ripple effect. A possible correction for this ripple effect could be based on its sensor-specific frequency (as suggested by Frank Wagner, DWD, personal communication, 2015), but is not addressed here.

Assuming the actual information content related to atmospheric backscatter is low above the ABL in the selected night-time profiles (i.e.  signal contribution is small cf. noise in the absence of clouds), the median climatology grouped by firmware plus transmitter configuration (Fig. 2, right) describes the  background signal composed of instrument-related background signal and potential cosmetic shift (i.e. $P^{bg}(r) = P^{bgi}(r) + P^{cs}(r)$). Although the range of values is large, IQR and median profiles have rather consistent statistics; shape of the background signal profile depends  on both sensor-individual hardware and firmware used. This is particularly evident when comparing  median night-time climatology profiles for  various configurations directly (Fig. 3).

The profiles for each sensor by firmware version (Fig. 3a) show that the complete background signal may be similar for sensors with the same generation of hardware (e.g. profiles of A & B both with CLE 311 + CLR311 are similar when running firmware 1.61 or 1.71; C & D are similar) however, this is not necessarily the case (e.g. background signal of S operating with CLE321 + CLR321 clearly differs from the background signal detected for C & D). Furthermore, the profile of the background signal may be altered by firmware. For sensors analysed here, profiles of the background signal are positive ($\sim 2\text{-}5 \cdot 10^{-14}$ arbitrary units (a.u.) at 2400 m) below 7000 m for firmware 1.61, generally decreasing with altitude range. The step change when profiles change sign is also evident in the climatology of the night-time profiles (time series in Fig. 2). For all background signal profiles observed with firmware 1.71, a strong, negative bias ($\sim 12\text{-}14 \cdot 10^{-14}$ a.u. at 2400 m) associated with the cosmetic shift applied, causes an overall negative background signal which increases (i.e. absolute values decrease) slightly with range below about 5500 m. Background signal profiles from newer hardware (sensors C and D, Fig. 2c, d) can have a similar shape independent of firmware (i.e. 2.01, 2.02, 2.03). Nocturnal profiles of sensor S (Fig. 2e) show less variability compared to the LUMO sensors, which is explained by the fact that block averages (30 s; 15 m) are analysed here instead of the high-resolution data (2 s; 5 m) initially acquired (Table 2, Sect. 2). Generally, the combination of individual hardware components and firmware used appears to determine the background, i.e. while sensors A and B are in good agreement for data gathered with versions 1.61 or 1.71, their backgrounds have opposite signs with version 1.72.

The seasonality evident in the time series (Fig. 2c, d, left) is related to the laser heat sink temperature which is used to further classify the background signal of sensors A-D into three sub-classes (Fig. 3 b, see legend). Profiles are only analysed above 2400 m as climatological measurements within the (sub-) urban ABL of London and Paris are inappropriate; if long-term measurements are available where ABL aerosol and moisture content are low (e.g. mountain sites), the climatology approach may provide valuable insights to much lower range gates.

To evaluate if the night-time climatology is a suitable basis  to assess the background signal profile, '' -measurements for four LUMO sensors with recent firmware and hardware configurations (Fig. 3c) are used under full-attenuation. The ceilometer window is covered by a Vaisala *termination hood* to mimic a clear night-time situation (i.e. supposedly no signal is backscattered to the receiver). Only internal contributions (e.g.  background signal) should contribute to the recorded signal. To eliminate  transient behaviour in the lowest range gates  the hood measurements are taken for 30 min periods. Later tests indicate observations at range < 50 m may require about 1 h to settle to a characteristic value (Fig. 4), in agreement with CeiLinEx CL51 ceilometer termination hood measurements (Frank Wagner, DWD, personal comm. 2015; www.ceilinex2015.de). While variations above this range do not show such a temporal drift, it is assumed values in the first four range gates in the initial termination hood profiles are significantly overestimated. Here, the profile is therefore set to be constant below the 5$^{th}$ range gate (Fig. 3c-e).

Average termination hood profiles are compared to night-time climatology profiles from the same laser heat sink temperature classes . For most sensors and firmware, the median night-time climatology agrees very well with the profile observed by the termination hood measurement (Fig. 3c). Only for ceilometer A (firmware 1.71) does the termination hood measurement have a slightly different shape, albeit with a similar order of magnitude. As there are no data available from the climatology approach for ranges below 2400 m, profiles are assumed to be constant up to this range. While this results in an obvious discrepancy between the climatology-derived background and the termination hood profiles (Fig. 3c), implications of this assumption are greatly reduced after range correction is performed (Fig. 3d-e). Although uncertainties remain regarding the profiles of background signal below a range of 2400 m, termination hood reference measurements give confidence that the night-time climatology measurements are not significantly influenced by backscatter from atmospheric particles and hence provide reasonable estimates of the background signal. This finding is extremely useful as it allows for the background signal of ceilometer sites that were operated in the past or that are difficult to access (e.g. termination hood measurements are unfeasible) to be evaluated based on the observed profile data alone.

Vaisala states (firmware release note) that no deliberate cosmetic shift is implemented in versions 1.72 and 2.03. Given that background signals from the earlier release versions are much closer to zero or even positive, it can be concluded that there is no (or negligible) cosmetic shift in versions 1.56, 1.61, 2.01, and 2.02 and the complete background signal $P^{bg}(r)$ is only composed of the instrument-related background signal $P^{bgi}(r)$. Of the versions tested, only firmware 1.71 profiles are shifted significantly towards negative values. The long-term estimates of hardware and firmware specific background signal $P^{bg}_{night}$ (instrument-related effects plus cosmetic shift; Fig. 3a) are used to determine an appropriate background correction derived for the 2400 – 7700 m range:

$$P^{bg}_{night}(r) = [P^{bgi}(r) + P^{cs}(r)]_{night}.$$  (2)

~~small with no distinct diurnal pattern, it is clearly affected by the response of the zero-level to background solar radiation for data collected with firmware version 1.71. The night time background profiles (Eq. (2)) determine the range-dependence of the background correction, while its magnitude further depends on $Z(t)$. The nocturnal average (4 h around midnight) of the signal at the top of the profile $\bar{Z}_{night}$ is subtracted so that the amplitude of $Z(t)$ only introduces diurnal variations and the background correction $P^B(r)$ remains close to the climatology $P^B_{night}(r)$ when solar radiation is absent. For firmware with no significant cosmetic shift, the background correction $P^B(r)$, describing the electronic background, can be determined by the night-time profiles. For firmware version 1.71 with strong cosmetic shift, profiles are offset by a diurnal pattern in the shape of the estimated zero-level:~~

For firmware with no significant cosmetic shift, the atmospheric contribution to the background correction is negligible so that a static correction over time can be applied defined by the night-time profiles

$$\sout{P^B(r) = \begin{cases} P^B_{night}(r) - \left(Z(t) - \bar{Z}_{night}\right), & firmware\ version = 1.71 \\ P^B_{night}(r), & firmware\ version \neq 1.71 \end{cases}} P^{bg}(r) = P^{bgi}(r) = P^{bg}_{night}(r). \tag{3}$$

As discussed, background profiles from sensors C and D have a small temperature dependence (Fig. 3b), however, the background signal of these sensors has an overall very small magnitude so that this thermal effect is considered negligible in the proposed correction (Eq. 3). Data with cosmetic shift (i.e. those collected with firmware 1.71) show strong diurnal variations in the signal background in response to background solar radiation. This indicates some contribution of the atmospheric background is retained in observations from this firmware version as the dynamic 'zero-bias-level' is effectively different from zero (Sect. 2). Because this is performed internally by the firmware, the exact contribution of the atmospheric background signal is not available for post-processing use. However, it can be approximated by the average signal $P^{top}(t)$ across the top range gates where the contributions from aerosol scattering to the signal can be deemed negligible. The calculation of $P^{top}(t)$ follows the approach taken to estimate the noise-floor $F(t)$ (i.e. cirrus clouds are masked out, Sect. 4.2). Only for data affected by the cosmetic shift (i.e. firmware 1.71) does $P^{top}(t)$ show significant values with a clear diurnal pattern that define the temporal variations of the background while the night-time background profiles (Eq. (2)) determine its range-dependence. To ensure the background correction $P^{bg}(r)$ remains close to the climatology $P^{bg}_{night}(r)$ when solar radiation is absent a nocturnal average $P^{top}_{night}$ (mean $P^{top}(t)$ of 4 h around midnight calculated for each day to be corrected) is subtracted:

$$P^{bg}(r) = P^{bg}_{night}(r) - \left(P^{top}(t) - P^{top}_{night}\right). \tag{4}$$

The derived background correction  $P^{bg}(r)$ (according to Eq. 4 for firmware 1.71 and Eq. 3 for other versions tested) can be applied in the post-processing to estimate the entirely background-corrected signal $\hat{P}$ without effects of cosmetic shift from the data recorded (Eq. (1)). This correction reduces the range-dependence of the observed signal so that the range-histograms of $\hat{P}$  $\cdot r^2$ (Fig. 1c) are more symmetric around zero than those of  ($P \cdot r^2$ (i.e. RCS, Fig. 1b) in all range gates in the free atmosphere, i.e. the median profile is close to zero.

All ceilometers tested here have a non-zero background profile, which confirms analysis by the Met Office (termination hood measurements and case study analysis) giving a negative background for other CL31 sensors in their network (Mariana Adam, Met Office, personal comm., 2014-2015). This creates additional challenges when deriving the aerosol backscatter coefficient from such measurements (M. Adam, Met Office, personal comm., 2015). For firmware versions without (or negligible) cosmetic shift , the background signal consists solely of the instrument-related contributions which may be small (on average $< |5 \cdot 10^{-14}|$ a.u.). Implications of these instrument specific variations might be limited for observations within clouds or in the ABL, where backscatter values tend to be large and mostly positive. However, the instrument-related background signal can reach significant values that may dominate any signal differences expected at the top of the ABL . The cosmetic shift in version 1.71 clearly affects observations within the ABL (Sect. 4.2). Note that the cosmetic shift and instrument-related background signal should be carefully evaluated before using noise for quality control purposes including absolute calibration and *SNR* calculations (Sect. 4).

**3.2. Range correction**

For a given concentration of atmospheric scatterers (cloud, aerosol, molecules) the strength of the backscattered signal returned to the ceilometer telescope and detector decreases by the square of the range $r$. Therefore, to relate scattering coefficients at different ranges, the signal $P$ is multiplied by $r^2$ at each range gate to obtain the range-corrected signal:

$$\tilde{P}(r) = \hat{P}(r)RCS(r) = P(r) \cdot r^2. \tag{5}$$

The signal $P$ is determined from the range-corrected signal reported by the CL31 (there termed 'range and sensitivity normalised attenuated backscatter') by reverting Eq. (5). Vaisala instruments have an option for the range correction to be applied only to the signal in the lower part of the profile up to a set range $r_{H2}$, where it is implicitly assumed that most of the data at further ranges consists of noise (setting: '*Message profile noise_h2 off*'). If no clouds are present in the profile, the raw signal is multiplied by a constant, height-invariant scale factor $k_{H2}$ above $r_{H2}$ (CL31: $r_{H2} = 2400$ m and $k_{H2} = r_{H2}^2 = 2400^2$). The partly range corrected signal reported $RCS_{H2}$ has two segments:

$$\tilde{P}_{H2}^{raw} = \begin{cases} P^{raw}(r) \cdot k_{H2}, & r > r_{H2} \\ P^{raw}(r) \cdot r^2, & r \le r_{H2} \end{cases} RCS_{H2} = \begin{cases} P(r) \cdot k_{H2}, & r > r_{H2}, \\ P(r) \cdot r^2, & r \le r_{H2}. \end{cases} \tag{6}$$

When clouds are detected, the cloud signal is range-corrected using Eq. (5), for range gates where cloud is determined to exist. To create a fully range-corrected signal from such observations for the whole vertical profile (according to Eq. (5), i.e. as if run with the setting '*Message profile noise_h2 on*') in the absence of clouds, the scale factor needs to be reversed and the range correction applied to the observations above $r_{H2}$:

$$\tilde{p}^{raw} = \begin{cases} \tilde{p}^{raw}_{H2}(r)/k_{H2} \cdot r^2, & r > r_{H2} \\ \tilde{p}^{raw}_{H2}(r), & r \le r_{H2} \end{cases} \quad RCS = \begin{cases} RCS_{H2}(r)/k_{H2} \cdot r^2, & r > r_{H2}, \\ RCS_{H2}(r), & r \le r_{H2}. \end{cases} \tag{7}$$

Still, this correction may only be applied where no clouds are present. Hence attenuated backscatter observations obtained with the setting '*Message profile noise_h2 off*' are of limited use (M Adam, Met Office, personal comm., 2014; www.ceilinex2015.de). For ceilometers operating with '*Message profile noise_h2 on*', all firmware applies the range correction throughout the entire profile and no constant scale factor is incorporated in this processing step. Hence it is recommended to operate with this setting turned on.

 The range-histograms of the range-corrected signal (Fig. 1b, c) illustrate the increase in signal variability with range. After applying the full range correction (Eq. (7)) to observations from a CL31 operated with '*Message profile noise_h2 off*' (rightmost panel in Fig. 1), the variability of the  signal is height-invariant above the ABL (Fig. 1a) while the expected increase is found in the range corrected signal (Fig. 1 b, c), i.e. it has the same signature as if it was recorded with the setting switched on.
* * *
**3.3. Optical overlap**

The receiver field of view reaches complete optical overlap with the emitted laser beam at a certain distance above the instrument. This overlap depends on instrument design. Overlap correction functions can be applied to partly account for this effect, with dimensionless multiplication factors determined empirically (e.g. Campbell et al., 2002). The overlap correction may either be performed by firmware or during post-processing. Uncertainty remains for observations at the closest range gates (e.g. Vande Hey, 2014; Hervo et al., 2016).

Applying an optical overlap correction $O(r)$ to the signal, yields the overlap-corrected signal:

$$P^{OC}(r) = P(r)/O(r). \tag{8}$$

Vaisala ceilometers have a single-lens, coaxial beam setup (Münkel et al., 2009). For the CL31, complete optical overlap is reached at about 70 m from the instrument (Fig. 5) and an overlap correction is performed by the firmware (i.e. $P(r) = P^{OC}(r)$). No other commercially available ceilometer offers complete overlap that close to the instrument. Vaisala overlap functions are verified both by ray tracing simulations and laboratory measurements.

**3.4. Near-range correction**

Although Vaisala suggests that the attenuated backscatter profile is reliable down to the first range gate, Sokół et al. (2014) document a distinct local minimum in CL31 attenuated backscatter observations at the 4th range gate persisting throughout their whole observational campaign.  As others have found artefacts in CL31 profiles below 70 m (e.g. Martucci et al. 2010; Tsaknakis et al. 2011) these lowest layers are often excluded during processing. As noted, van der Kamp (2008) smoothed out systematic features by strong vertical averaging, but this removes the possibility of identifying any atmospheric features close to the surface. Without correction, these features may cause detection of significant gradients when examining profiles to diagnose mixing heights or top of the ABL. Artefacts in the first 70 m could be related to the incomplete optical overlap (Sect. 1) but are more likely associated with a hardware-related perturbation and a correction introduced by Vaisala to prevent unrealistically high values in the near-range when the window is obstructed.
* * *
~~The receiver field of view reaches complete optical overlap with the emitted laser beam at a certain distance above the instrument. This overlap depends on instrument design. Overlap correction functions can be applied to partly account for this effect. They are dimensionless multiplication factors (0 (nearest the instrument) to 1 (range gates above the point of complete overlap)) which are determined empirically (e.g. Campbell et al., 2002). The overlap correction may either be performed by firmware or during post-processing. Uncertainty remains for observations at the closest range gates (e.g. Martucci et al., 2010; Sokół et al., 2014; Vande Hey, 2014).~~

$$\hat{P}^{oc}(r) = \frac{\hat{P}(r)}{O(r)} \quad\quad (7)$$
* * *
Given the primary function of cloud base height detection, Vaisala  CL31 firmware addresses effects causing extremely high backscatter values outside of clouds. Under severe window obstruction (e.g. leaf on  window), values in the first range gates can be unrealistically high. A correction is applied to restrict the backscatter profile in the  ranges closest to the instrument. At times, this correction introduces extremely small values at ranges <

50 m that are clearly offset from the observations above this height. In addition to this artefact from the obstruction correction, for some sensors, backscatter values in the range of 50-80 m are slightly offset by a hardware-related perturbation. Both  artefacts from the obstruction correction and  hardware-related perturbations do not impact cloud detection , vertical visibility or boundary layer structures (> 80 m). It is only for attenuated backscatter closer than 80 m that

10 these artefacts needs to be accounted for. The issues are not firmware-specific, apart from versions 1.72 and 2.03 in which the artefacts of obstruction-correction and hardware-related perturbation have been mostly removed. These low-level artefacts are expected to be consistent in time  from older firmware.

To evaluate the effect of the obstruction-correction and hardware-related perturbation, profiles of the range-corrected

15 reported signal in the lowest 90 m are normalised by the value at 100 m ($\tilde{\beta}^{OC}$$RCS$(n)/$\tilde{\beta}^{OC}$$RCS$(10), with n = range gate. ; here using LUMO sensors A - D with range resolution of 10 m, Table 2). Selecting daytime profiles 11-16 UTC $\tilde{\beta}^{OC}$, $RCS$ < 200 × 10$^{-8}$ a.u.. for range < 400 m; note no absolute calibration is applied) shows the normalised profiles have a consistent shape across the four LUMO CL31 sensors (Fig. 6a). The median profile has a small reduction in backscatter  at 80 m

20 (8$^{th}$ gate), a distinct peak at 50 m (5$^{th}$ gate) and rather similar values in the lowest four gates (< 40 m). The artefacts are of smaller magnitude in observations from sensor B . The normalised values in the first four range gates  have two different regimes: while for most profiles the normalised overlap- and range-corrected signal ranges between 1.0 and 1.2 (Fig. 6b), a small fraction of samples have lower values ($\tilde{\beta}^{OC}$$\tilde{\beta}^{OC}$$RCS$(2)/$RCS$(10) < 0.8; Fig. 6c). This effect is likely explained

25 as an artefact of the obstruction correction while the deviations at 50 m and 80 m are associated with the hardware-related perturbation. The observed range provides uncertainty information for the detection of the near-range artefacts; the peak at the 5$^{th}$ range gate indicates an overestimation of about 40-50% while systematic differences are commonly < 20% for the remaining range gates below 100 m. For dry and well-mixed conditions, profiles observed with firmware 1.72 or 2.03 (Fig. 6d)  the  obstruction correction and hardware-related

30 perturbation are removed with these updates.

Based on the median climatological profiles (Fig. 6

), a near-range correction is proposed to reduce the impact of the obstruction correction and hardware-related perturbation. Only profiles that roughly match the general shape of the climatology are corrected, i.e. if strong vertical gradients in the signal are observed (e.g. descending fog) the near-range correction is inapplicable. However, these conditions are usually small compared to the physical processes influencing the attenuated backscatter across the profile.

Given all sensors tested have a distinct peak at a certain range gate (Fig. 6a) this peak is used to indicate if a correction should be applied. The inverse approach could correct observations with a strong local minimum at the 4th range gate rather than a peak as reported by Sokół et al. (2014). The aim is to apply the near-range correction only to profiles with a pronounced peak value that appears physically unreasonable. First, the range gate with the peak is identified from the climatology (5th range gate for LUMO sensors). Second, the peak strength is defined as the ratio of the range-corrected signal reported at this range gate to that reported at the adjacent gates (i.e. 4th and 6th for LUMO sensors). If both these peak-strength indicators of a given profile are at least 25% as strong as the peak-strength indicators of the climatology profile, the values of this profile in the near-range (< 100 m) are divided by the median climatology profile (Fig. 6b). Profiles affected by the obstruction correction, i.e. with clearly offset values in the first four range gates, are treated separately. If the first peak-strength indicator (i.e. the one below the peak) is at least 50% as strong as the respective indicator of the climatology of this regime (Fig. 6c) and the value at the range gate of the peak is greater than the values in the two range gates above, the respective median climatology profile is used for the correction (Fig. 6c).

Correction functions can help to reduce the processing artefact due to the obstruction correction and the hardware-related offset  LUMO ceilometers A-D, see Table 2). Observations taken with firmware versions < 1.72 (for systems running with engine board plus receiver combination CLE311 + CLR311) or < 2.03 (CLE321 + CLR321) have clear near-range effects (Fig. 6a-c) evident in data recorded (Fig. 7i, iii, v). Two examples are clearly affected by the obstruction correction (Fig. 7iii, ceilometer C and D) with values in the lowest four range gates negatively offset. After the near-range correction is applied, this effect is reduced and the artificial peaks at the 5th range gate are mostly removed (Fig. 7ii, iv, vi). Although some residual effects may remain, extreme vertical gradients encountered within the lowest 100 m of the original range corrected signal reported by the CL31 ceilometers are mostly removed.

Vaisala introduced a correction for the near-range artefacts that proves efficient in dry conditions (Fig. 6d), however, if attenuation is increased due to hygroscopic growth, the peak at the 5th range gate is still evident in the normalised *RCS* profile (Fig. 7vii). Applying the near-range correction proposed for observations from earlier firmware version (as used for Fig. 7ii, iv, vi), the artefacts could still be removed (Fig. 7viii, ceilometer A and B), however, it could also result in an over-correction (Fig. 7viii, ceilometer C and D). Note that this approach can only be tested on sensors for which a historic dataset of measurements with older firmware versions is available to calculate the respective correction profiles (Fig. 6). Given the near-range correction introduced by Vaisala in version 1.72 and 2.03 is not sufficient in moist conditions with gradients along the profile (Fig. 7vii) it was proposed to Vaisala to remove their correction again so that the near-range correction can be applied during post-processing.

**5.4. Absolute backscatter and quality assurance**

**4.1. Absolute calibration**

The range-corrected attenuated backscatter $\beta\beta \cdot r^2$ describes the range-corrected, entirely and background-corrected and overlap-corrected signal calibrated by the lidar constant $C$:

$$\beta = \tilde{P}^{oc}/C \, \beta \cdot r^2 = \hat{P} \cdot r^2/C. \qquad (9)$$

The range correction can be reversed for the attenuated backscatter to yield the entirely background-corrected and overlap-corrected attenuated backscatter $\hat{\beta}$:

$$\hat{\beta} = \beta/r^2. \qquad (9)$$

The lidar constant $C$ is a function of the range-independent parameters of the lidar equation, including the speed of light, the area of the receiver telescope, the temporal length of a laser pulse, a system efficiency term and the mean laser power per pulse (Weitkamp, 2005). It depends on the instrument receiver design and its laser. When the instrument is new, system efficiency and laser power are high. At this stage, the lidar constant for internal calibration is determined by a factory-based test ($C = C_{factory}$). Even with regular cleaning and maintenance, the performance of a sensor changes over time (e.g. aging of the laser, changes in window transmissivity). To account for such possible variations in laser output and detector capability over time, the ceilometer firmware monitors the laser output energy and determines a relative calibration-correction factor $c_{monitor}(t)$ which is a time-specific lidar constant applied internally:

$$C_{internal}(t) = C_{factory} \cdot c_{monitor}(t). \qquad (10)$$

Over time, this internal checking of the instrument performance potentially provides a continuous relative calibration. Given that the signal output by the ceilometer already has the internal calibration applied, it is labelled 'attenuated backscatter' by

the manufacturer. However, it has been shown that the internal calibration factor $C_{internal}$ does not always fully represent the actual lidar constant (e.g. O'Connor et al., 2004) and that an absolute calibration should be performed in sufficiently known atmospheric conditions. Given the background noise of the CL31 sensors dominates over the molecular backscatter (Sect. 2), the stratocumulus cloud technique (O'Connor et al., 2004) is the most appropriate calibration technique for the Vaisala sensors. This agrees with the findings of the TOPROF community (Maxime Hervo, Meteo Swiss, personal comm., 2015). The stratocumulus cloud technique relates the observed signal to the known integrated attenuated backscatter coefficient associated with thick liquid clouds. This absolute calibration technique is applied externally, i.e. as part of the post-processing:

$$\beta = \left(\frac{\tilde{P}^{OC}_{internal}}{C_{internal}(t)}\right)/c_{absolute}(t)\cdot\beta \cdot r^2 = \left(\frac{P_{internal}}{C_{internal}(t)}\right)/c_{absolute}(t) \cdot r^2 \tag{11}$$

The absolute calibration coefficient $c_{absolute}(t)$ may be constant in time $c_{absolute}(t) = c_{absolute}$ (Hopkin et al., 2016). A laser at the CL31 operating wavelength (~ 905 nm) is sensitive to absorption of water vapour in the atmosphere which can have implications for the absolute calibration (Markowicz et al., 2008; Wiegner and Gasteiger, 2015). As  evaluation of absolute calibration techniques is beyond the scope of this study, for simplicity the impact of this external calibration is neglected (i.e. $c_{absolute}$ $(t) = 1$).

**4.2.  Signal strength and noise**

Given that noise is a critical component of the attenuated backscatter recorded, data with values below a certain signal-to-noise ratio (*SNR*) are unlikely to contain sufficient information about the state of the atmosphere. Where high-resolution observations are obtained, rolling spatial (along-range) and temporal averaging increases the signal contribution relative to the noise. For every range gate $r$ and time step $t$, the smoothed attenuated backscatter is the average over a temporal window of fixed size $2 w_t + 1$ (with $w_t$ time steps) and a range window of fixed size $2 w_r + 1$:

$$\hat{\beta}_{smooth}\beta_{smooth}(t,r) = (2w_t + 1)^{-1} \cdot (2w_r + 1)^{-1} \sum_{k=r-w_r}^{k=r+w_r}\sum_{h=t-w_t}^{h=t+w_t}\hat{\beta}(h,k)\cdot\sum_{k=r-w_r}^{k=r+w_r}\sum_{h=t-w_t}^{h=t+w_t}\beta(h,k) \tag{12}$$

~~Optimal window length depends on hardware characteristics (i.e. noise-levels), resolution settings for raw data acquisition and the application. Here window lengths combining to an averaging factor of about 1000 have been found suitable to prepare data for the detection of mixing height with a relatively larger temporal averaging window (i.e. $w_t = 50$; $w_r = 5$) as features of the ABL structure show more variability in the vertical. Such large window sizes can significantly improve the SNR, i.e. signal strength compared to average background noise. Still, Sokół et al. (2014) suggest temporal averaging should be shorter in the morning transition period when boundary layer dynamics may vary extensively over time scales of 30 – 60 min. They use a 5 min averaging window on the data prior to their mixing height analysis.~~

Optimal window length depends on hardware characteristics (i.e. noise-levels), resolution settings for raw data acquisition and the application. Here, window lengths combining to a total of about 1000 have been found suitable to

prepare data for the detection of mixing height with a relatively larger temporal averaging window (i.e. $w_t = 50$; $w_r = 5$ which equals 25.25 min and 110 m for the LUMO sensors, see Table 2) as features of the ABL structure show more variability in the vertical than over time. Such large window sizes can significantly improve the *SNR*, i.e. signal strength compared to average background noise. Sokół et al. (2014) suggest temporal averaging should be shorter (e.g. 5 min averaging window prior to mixing height analysis) in the morning transition period as boundary layer dynamics may vary at 30 – 60 min time scales. To significantly increase *SNR*, Stachlewska et al. (2012) use Gaussian smoothing (Jenoptik CHM15K ceilometer) with linearly increasing range window-widths, however, this is can result in extensive computing time. *BLview* (Vaisala's boundary layer detection software, e.g. used by Tang et al., 2016) has range-variant smoothing windows.

The quality of range-corrected attenuated backscatter  can be evaluated by comparison to the *noise floor*. The latter represents variations associated with electronic and optical noise and noise introduced by the solar background light. If no high cirrus clouds are present, it is assumed the signal from the very highest range gates contains only noise (i.e.  atmospheric signal contribution is negligible). In this case, the noise floor $F$ can be defined as the mean $\bar{\beta}$ plus standard deviation $\sigma_\beta$ of the  attenuated backscatter $\hat{\beta}$  $\beta$ (i.e. before range correction) across a certain number of gates from the top of the profile. Statistics are applied across these gates at the top of the profile and moving temporal windows (as in Eq. (12)):

$$F(t) = \bar{\beta}(t) + \sigma_\beta(t). \tag{13}$$

Here, the highest 300 m of the profile ($N = 30$ at 10 m resolution) are used to determine the noise floor to ensure sufficient representation of the range-variability. Similar results are obtained with slightly more range gates . The discontinuity and increased noise levels around 7000 m (Sect. 3.1) makes it unadvisable to include more than  600 m to calculate the noise floor. The mean $\bar{\beta}$ across the top range gates is usually small and fluctuates around zero. However, if the background correction  (Sect. 3.1) is not performed it can have a slight offset from zero and even a diurnal pattern for data acquired with firmware version 1.71 which performs the cosmetic shift based on the dynamic zero-bias-level (Sect. 2). Calculated from the entirely background--corrected signal or attenuated backscatter (see Eq. (1-4)), the noise floor $F$ is nearly equal to the standard deviation $\sigma_\beta$ across the top range gates.

To ensure that profiles used for the calculation of $F$ do not contain any cirrus clouds, which can provide significant backscatter even  at the furthest ranges of the profile, the "relative variance" $RV(t, r)$ (or coefficient of variation) is used to mask cloud observations (Manninen et al., 2016). For each time $t$ and range $r$ (at the top of the profile), the relative variance is the ratio of the standard deviation $\sigma_\beta(t, r)$ to the mean $\bar{\beta}(t, r)$, with statistics applied over moving windows (as in Eq. (12)), along range and time (here, $w_r = w_t = 3$ were

used):

$$RV(t,r) = \left(\frac{\sigma_\beta(t,r)}{\overline{\beta}(t,r)}\right)^2. \tag{14}$$

If $RV(t,r)$ is sufficiently small, then the backscatter is interpreted as a true signal backscattered by atmospheric constituents of interest (e.g. cirrus clouds). Such backscatter values should not be incorporated into the calculation of the noise floor (Eq. (13)). Rather $F$ should be estimated for observations when $RV$ exceeds a threshold $T_1$, $\hat{\beta} = \hat{\beta}(\beta = \beta(RV > T_1)$. A

5    threshold of $T_1 = 1$ indicates that the variability exceeds the mean signal and can be used to mask strong backscatter from clouds. Times with a small number of gates (e.g. layer  $\leq 100$ m) available for calculation of the noise floor (i.e. when many of the top range gates have cirrus) can be interpolated linearly in time. A day with cirrus in the top gates (Fig. 8) illustrates how the threshold $T_1$ can be applied to convert the $RV$ field (Fig. 8a) into a mask to remove the cirrus signal from the attenuated backscatter (Fig. 8b). Based on the attenuated backscatter with clouds masked out (Fig.

10    8c), the noise floor $F$ is calculated over the course of the day (Fig. 8d) and the area missing due to the presence of cirrus is interpolated linearly over the time period where the attenuated backscatter has been masked out.

The $SNR$ is calculated from smoothed, non range-corrected attenuated backscatter (Eq. (12)) and the noise floor $F$:

$$SNR(r,t) = \frac{\hat{\beta}_{smooth}(r,t)\,\beta_{smooth}(r,t)}{F(t)\;F(t)}. \tag{15}$$

Note that in clean air with low aerosol content the dominant scattering is molecular which is below

15    the sensitivity of a CL31 ceilometer (Sect. 2). Furthermore, thick liquid clouds have the ability to (almost) fully attenuate the ceilometer signal so that any returns from above such a cloud layer (or even within it) correspond to noise rather than to atmospheric scattering from  particles or molecules. Hence, at certain heights the information content of the signal may be limited. To evaluate where the signal contribution is clearly distinguishable from the noise, Welch's t-test (Welch, 1947) is performed comparing the distributions of $\hat{\beta}_{smooth}\beta_{smooth}$ and $F$, assuming they are both normally

20    distributed. As $\hat{\beta}_{smooth}\beta_{smooth}$ was found to deviate from normality below a range of about 500 m, the t-test was only performed for higher ranges. A p-value $< 0.01$ was chosen as accepting that $\hat{\beta}_{smooth}\beta_{smooth}$ significantly exceeds the noise floor at the respective time step and range gate. The acceptance level calculated for each $SNR$ bin (Fig. 9) reveals a clear divide between observations with high information content and those with a magnitude comparable to the noise floor (or lower). For the LUMO sensors (Table 2)

25    acceptance levels of 50% - 90% correspond to $SNR$ values of $0.05 – 0.20$, which indicates the range of threshold values that can be selected depending on whether a more relaxed or conservative filtering is desired. These low values can be explained by the fact that most observations with no significant signal contribution become or remain negative after  smoothing (Eq. (12)) has been applied.

The impact of averaging, background correction and noise filtering on observations taken by different sensors and firmware versions is illustrated based on three case study days  (Fig. 10): 24 July 2012 with clear-sky conditions comparing sensor A running with firmware version 1.61 (row 1) and to sensor C with firmware 2.02 (row 2), 22 June 2012 with some boundary layer clouds present comparing sensor B with 1.71 (row 3) and sensor C with firmware 2.02 (row 4), and 29 June 2015 with a few isolated medium- and high-level clouds showing observations from sensor S with 2.01 (row 5). The range-corrected attenuated backscatter reported (Fig. 10a) is quite noisy in all sensor observations and  the evolution of the ABL is difficult to discern. When the moving average is applied (Fig. 10a, b). ~~The increase in signal due to the moving average is evident in both old and new generation sensors running with different firmware versions. The impact of the background correction (Eq. (1)-(3)) has serious implications for observations in the ABL given that the electronic background might obscure the transition between the boundary layer top and the clear atmosphere above (compare Fig. 9b, c, firmware 1.61) or the cosmetic shift might reduce the information content so strongly that certain regions within the boundary layer are lost (compare Fig. 9b, c, firmware 1.71). Here, the background correction can increase data availability. The impact of the background correction is small for new-generation sensors running with versions 2.xx. Applying the SNR filter based on the statistical threshold helps to distinguish data with significant information content (compare Fig. 9c, d) so that quality can be assured for later applications (e.g. mixing height detection). Data quality of sensors running with new engine boards and versions 2.xx are clearly superior to older generations~~b), the signal contribution clearly increases so that aerosol layers can be identified visually. However the contrast between ABL and the clear air above varies greatly with sensor and firmware version. While the ABL reveals distinct attenuated backscatter signatures for data from sensor C with firmware 2.02 (rows 2 and 4), the values in the free troposphere are elevated for sensors A (row 1) and S (row 5). This is explained by the different profiles of background signal inherent in these observations (Fig. 3a): sensor C has a small and slightly negative background signal which barely affects observations above the ABL while both sensor A (firmware 1.61) and sensor S (2.01) have a positive background signal leading to an overestimation of signal below about 5000 m. Even more severe is the impact of the cosmetic shift inherent in observations from sensor B (1.71; row 3), which reduces the signal significantly even within the ABL. For the example shown, average values become negative below the boundary layer clouds so that no mixing height detection algorithm would be able to derive relevant statistics.

The described artefacts can mostly be accounted for by the proposed background correction (Eq. (1)-(4); Fig. 10c). While it can help to improve the contrast at the boundary layer top for sensors A (row 1) and S (row 5) it can revert the cosmetic shift in data from sensor B (row 3). In the latter case, the background correction can increase data availability. It should be noted that the systematic ripple effect (sensors B and C; see Sect. 3.1) becomes apparent after the background correction. Although the ripple is somewhat coherent and not truly random, affected areas above the ABL can still be successfully masked by the SNR filter (Fig. 10d). For all sensors, this statistical threshold helps to distinguish data with significant information content

(compare Fig. 10c, d) so that quality can be assured for later applications (e.g. mixing height detection). Still, some significant noise may remain near the ABL top for the older generation of hardware running with firmware 1.xx (row 1 and 3). It can be concluded that data quality of sensors of the recent hardware generation, i.e. those operating firmware versions 2.xx (here sensors C and S) are clearly superior to older generations (sensors A and B).

**6.5. Summary**

Ceilometers are valuable instruments with which to study not only clouds, but also the ABL and elevated layers of aerosols. Vaisala CL31 sensors provide good quality attenuated backscatter (as it is not absolutely calibrated, here it is referred. While their cloud base height product might be readily useful, to as 'raw signal'). However, to use these understand the profiles of attenuated backscatter the user needs to be aware of the instrument model's specific hardware and firmware. The following sections summarise aspects are useful to consider in post-processing of CL31 ceilometer attenuated backscatter profiles:.

By taking into account these instrument-specific aspects of the CL31 profile observations, data quality and availability can be improved. If data are collected according to best practice, as recommended above and issues are being corrected for in the post-processing (e.g. applying the proposed methods) and sensors are carefully calibrated, then the attenuated backscatter observations might prove useful for NWP model verification and evaluation, and potentially even for data assimilation.

**5.1. Instrument-specific characteristics and issues**

- Initial, internal averaging of the sampled ceilometer signal is applied over two selected time intervals that depend on the range and the user-defined reporting interval and the range for firmware versions < 1.72, 2.01 and 2.02. Data acquired with firmware 1.72 or 2.03 are more consistent than earlier versions because the whole profile (at all range gates) is treated equally with an internal averaging interval of 30 s.

- If the user-defined reporting interval is shorter than 30 s, consecutive profiles partly overlap in time and are hence not completely independent.

- When averaging several profiles, a discontinuity is evident at around both 4940 m and 7000 m for all sensors and firmware versions. These regions of increased noise are introduced by the data storage procedure of the firmware which slightly changes its operating mode after a certain number of gates have been collected. Care should be taken when looking at gradients or statistics near these heights.

- Depending on firmware version, a 'cosmetic shift' is applied to the attenuated backscatter profiles. This shift should be reversed before using any part of the entire profile for analysis. TheOf the firmware tested, the cosmetic shift appears to be negligible for all versions except for version 1.71 in which a strong negative shift is applied to the observations.

- In addition, a range-dependent electronic, instrument-related background signal is inherent in the recorded signal and this altersreported, altering the profiles systematically. The background (electronic noisesignal values (instrumentrelated background signal plus potential cosmetic shift) tend to be either predominantly positive or predominantly negative in ranges below 6000 m and to switch sign between about 6000 – 7000 m.

- ~~A climatology of night time profiles is used to determine the background correction that is required. A comparison with dark current measurements using a termination hood proves the nocturnal climatology accurately describes the background profile. Thus, the two can be considered equivalent and the background correction can be determined through either termination hood reference measurement (e.g. if profile observations are not available for a long time) or the climatology approach (e.g. if using historical data or the ceilometer site is difficult to access). Neither technique provides reliable information below about 2400 m so the profile is assumed to be range-invariant in this region.~~

-

- Both the range-dependent instrument-related background signal and the cosmetic shift applied may cause issues for studying the ABL because signal differences expected at the ABL top may be obliterated or the signal reduced too strongly for successful mixing height detection.

- Molecular scattering at the instrument wavelength is very weak, typically below the sensitivity of the instrument.

- The CL31 measurement design accounts for temporal variations in solar radiation by introducing a variable zero-bias-level so that the atmospheric background signal is inherently accounted for in the signal reported. However, solar radiation still contributes to the random noise.

- In the absence of cloud, rain or elevated aerosol layers, the recorded signal includes the instrument-related background signal plus potential cosmetic shift and noise associated with both the instrument (electronics and optics) and the solar background radiation. Instrument-related background signal and cosmetic shift should be carefully evaluated before using noise for quality control purposes.

- For some instruments, a 'ripple' effect is detected that superimposes a wave-type structure over the random noise. For the sensors evaluated, this was found in two generations of the engine board plus receiver (CLE311 + CLR311 and CLE321 + CLR321), but only for transmitter type CLT321. Temporal rolling averages may enhance this ripple effect at short time scales usually used for smoothing raw attenuated backscatter observations (i.e. minutes to hours ). Further investigation indicates that the ripple shows some response to the level of attenuation in the ABL.

- Vaisala instruments have a setting ('*Message profile noise_h2 off*') that restricts the range correction to the signal in the lower part of the profile up to a set, critical range. It is implicitly assumed that most data at ranges beyond

this critical height contain only noise. If no clouds are present in the profile, then the signal is simply multiplied by a constant, height-invariant scale factor. Where clouds are detected, the signal is actually range-corrected as usual, but only for range gates where cloud is determined to exist.

- Several artefacts may be found in the lowest range gates close to the instrument. The co-axial beam design of the CL31 ceilometer allows  complete overlap to be reached at 70 m. Below this range, an overlap correction is applied internally by the sensor.

- In addition to the overlap correction, Vaisala applies another correction to observations from the first few range gates to avoid exceptionally high readings when the ceilometer's view is obstructed (e.g.  a leaf on the window,). At times, the *obstruction correction* introduces extremely small values at ranges < 50 m that appear unrealistically offset from the observations above this height. Attenuated backscatter values may also be slightly offset in the range of 50-80 m which can be explained by a hardware-related perturbation.

- Although CL31 output is labelled as attenuated, range-corrected backscatter, the absolute calibration might not be accurate enough for use in meteorological research. The stratocumulus or liquid cloud calibration (O'Connor et al., 2004) can be used to determine the instrument-specific lidar constant based on external properties. This allows absolute calibration to be performed during post-processing. Data gathered with instruments that cause a strong electronic background signal and/or with firmware that applies the cosmetic shift (version 1.71), should be corrected for these effects before the calibration is applied. Note that absolute calibration is included for completeness, but is not addressed here.

**5.2. Proposed corrections**

- To create a fully range-corrected signal from data gathered with the setting '*Message profile noise_h2 off*' for the whole vertical profile (as if the setting '*Message profile noise_h2 on*' had been used) in the absence of clouds, the scale factor needs to be reversed and range correction applied to the observations above the critical height.

- A climatology of night-time profiles can be used to determine the background correction that is required to account for the instrument-related background signal and potential cosmetic shift. A comparison with termination hood measurements proves the nocturnal climatology accurately describes the background signal. Thus, the two can be considered equivalent and the background correction can be determined through either termination hood reference measurements (e.g. if profile observations are not available for a long time) or the climatology approach (e.g. if

historical data or ceilometer site access are difficult). No reliable information can be derived from the climatology technique below about 2400 m given the presence of the ABL. However, termination hood profiles show the magnitude of the instrument-related background signal below this range are rather small after range correction so the profile can be assumed range-invariant in this region.

- The cosmetic shift has some temporal variation through the day, indicating some influence of the atmospheric background is retained during internal processing. This effect can be accounted for in the background correction by including an offset based on average observations in the top range gates.

- The artefacts related to obstruction correction and hardware-related perturbation are mostly accounted for in versions 1.72 and 2.03, however, small effects remain under situations with considerable attenuated backscatter. Hence, removal of this correction in the next firmware update to allow for consistent corrections to be applied during post-processing was recommended to Vaisala. Data from earlier versions (and probably later versions) need to be corrected during post-processing if observations from the near range are to be analysed; a correction procedure has been proposed based on climatological statistics of well-mixed atmospheric profiles.

- Quality assuranceAs the noise in the background-corrected signal is independent of range, it can be determined using the top-most range gates in the profile where the contribution of real atmospheric scattering is negligible in the absence of high cirrus clouds. The *noise floor* is here, taken as the mean plus standard deviation of the background corrected signal (before range-correction is applied), is calculated across moving time windows in range and time over those top range gates with the mean generally negligible after background correction. Regions containing significant aerosol or cloud should be excluded. They are efficiently masked based on theirthe relative variance.

- To increase the signal strength,-to-noise ratio (*SNR*), a moving average is calculated for the non range-corrected attenuated backscatter across set windows in range and time. The relation of these smoothed statistics and the noise floor defines the signal-to-noise ratio (*SNR*). The latter*SNR* which may be used to mask observations where the noise exceeds the actual information content of atmospheric signal. A suitable *SNR* threshold to distinguish the signal from the noise region is estimated based on Welch's t-test.

- Data quality and *SNR* of sensors running with engine boards CLE321, receivers CLR321 and firmware versions 2.xx are clearly superior to those of the old generation (CLE311 + CLR311).

- In response to results presented here and discussions within the TOPROF community, Vaisala released two recent firmware versions: 1.72 for sensors running with older generation hardware (engine board CLE311 and receiver CLR311) and 2.03 for sensors running with newer generation hardware (engine board CLE321 and receiver CLR321). Data collected with these two firmware versions are more consistent and show great improvement in the attenuated backscatter profiles when compared to the data from older versions.firmware versions. Additional suggestions are

communicated to the manufacturer to allow for correction of the near-range artefacts during post-processing rather than performing it online.

**5.3. Concluding recommendations**

Assuming  the sensors evaluated in this study are representative of CL31 ceilometers in general, the following recommendations are made:

i. To operate CL31 sensors with the setting *Message profile noise_h2 on.*

ii. A reporting interval (temporal resolution) of at least 15 s is recommended (despite down to 2 s being possible).

iii. It is advised to operate CL31 sensors with engine board CLE321 + receiver CLR321 and firmware version 2.03.

iv. If only older hardware (CLE311 + CLR311) is available, firmware version 1.72 (or later) should be used.

v. The instrument-related background signal should be carefully evaluated for all sensors and firmware versions. This can be achieved based on night-time climatology statistics or termination hood measurements. Correction of the range-dependent background signal may improve the contrast between the ABL and the clearer air above.

vi. For data gathered with firmware  1.71,  the cosmetic shift  can be corrected based on a combination of background signal profile estimates and average attenuated backscatter across the profile top range gates in the absence of cirrus clouds.

vii. If information close to the sensor (< 100 m) is of interest near-range artefacts should be corrected in historical data collected with firmware versions 1.54, 1.61, 1.71, 2.01 or 2.02.This correction might generally not be necessary for data gathered with firmware 1.72 or 2.03, however, it was found to yield some improvement under moist conditions.

Given the impact of

viii. both hardware and firmware on attenuated backscatter profiles from  CL31 ceilometers, any publication of such data should clearly state relevant details on hardware generation and firmware versions used, if any changes  the setup  were made during the measurement period analysed, and post-processing undertaken.

**Acknowledgements**

This study was funded by H2020 URBANFLUXES, NERC ClearfLo (H003231/1), NERC Airpro, NERC//Belmont TRUC NE/L008971/1 G8MUREFU3FP-2201-075, European Cooperation in Science and Technology (COST) action 'TOPROF': ES1303, King's College London and University of Reading. For providing sites and other support we thank KCL Directorate of Estates and Facilities, ERG/LAQN and RGS/IBG; the many staff and students at University of Reading and KCL who contributed to data collection. Met Office, Meteo France, and the SIRTA observatory are kindly acknowledged for providing observations. We would like to acknowledge the useful discussions within the TOPROF community (especially with Mariana Adam, Met Office; Frank Wagner, DWD; and Maxime Hervo, Meteo Swiss). We thank all anonymous reviewers for their comments and helpful suggestions.

**Author contribution**

Simone Kotthaus is involved in maintaining the LUMO measurement network, performed all analysis, developed correction procedures and wrote the main parts of the manuscript. Ewan O'Connor was involved in the development of the background correction, the calculation of SNR and provided useful comments to the manuscript. Christoph Münkel provided useful information on sensor specifics and internal processing procedures, wrote some parts of the manuscript, and provided the TOPROF firmware versions tested in this study. Cristina Charlton-Perez provided CL31 observations from the Met Office sensor, contributed to writing the manuscript, was involved in discussions on sensor specifics, background correction, range correction and SNR calculations. Martial Haeffelin provided CL31 data from SIRTA, was involved in discussions on sensor specific, instrument-related background corrections, near-range corrections, *SNR* calculation, terminology and provided useful comments to the manuscript. Andrew M. Gabey developed the approach for calculating an appropriate SNR threshold. C. Sue B. Grimmond provided CL31 measurements from the LUMO measurement network, was involved in discussions on instrument specifics and all data processing aspects and provided useful comments to the manuscript.

**References**

Bromwich, D. H., Nicolas, J. P., Hines, K. M., Kay, J. E., Key, E. L., Lazzara, M. A., Lubin, D., McFarquhar, G. M., Gorodetskaya, I. V., Grosvenor, D. P., Lachlan-Cope, T. and van Lipzig, N. P. M.: Tropospheric clouds in Antarctica, Rev. Geophys., 50(1), RG1004, doi:10.1029/2011RG000363, 2012.

Campbell, J. R., Hlavka, D. L., Welton, E. J., Flynn, C. J., Turner, D. D., Spinhirne, J. D., Scott, V. S. and Hwang, I. H.: Full-Time, Eye-Safe Cloud and Aerosol Lidar Observation at Atmospheric Radiation Measurement Program Sites: Instruments and Data Processing, J. Atmos. Ocean. Technol., 19(4), 431–442, doi:10.1175/1520-0426(2002)019<0431:FTESCA>2.0.CO;2, 2002.

Cao, N., Zhu, C., Kai, Y. and Yan, P.: A method of background noise reduction in lidar data, Appl. Phys. B, 113(1), 115–123, doi:10.1007/s00340-013-5447-9, 2013.

de Haij, M., Wauben, W. and Klein Baltink, H.: Determination of mixing layer height from ceilometer backscatter profiles, edited by J. R. Slusser, K. Schäfer, and A. Comerón, Remote Sens., 63620R–63620R–12, doi:10.1117/12.691050, 2006.

Emeis, S.: Surface-Based Remote Sensing of the Atmospheric Boundary Layer, Springer Science & Business Media, 2010.

Emeis, S., Schäfer, K. and Münkel, C.: Surface-based remote sensing of the mixing-layer height – a review, Meteorol. Zeitschrift, 17(5), 621–630, doi:10.1127/0941-2948/2008/0312, 2008.

Emeis, S., Schäfer, K. and Münkel, C.: Observation of the structure of the urban boundary layer with different ceilometers and validation by RASS data, Meteorol. Z., 18(2), 149–154, doi:10.1127/0941-2948/2009/0365, 2009.

Emeis, S., Schäfer, K., Münkel, C., Friedl, R. and Suppan, P.: Evaluation of the Interpretation of Ceilometer Data with RASS and Radiosonde Data, Boundary-Layer Meteorol., 143(1), 25–35, doi:10.1007/s10546-011-9604-6, 2011a.

Emeis, S., Forkel, R., Junkermann, W., Schäfer, K., Flentje, H., Gilge, S., Fricke, W., Wiegner, M., Freudenthaler, V., Groβ, S., Ries, L., Meinhardt, F., Birmili, W., Münkel, C., Obleitner, F. and Suppan, P.: Measurement and simulation of the 16/17 April 2010 Eyjafjallajökull volcanic ash layer dispersion in the northern Alpine region, Atmos. Chem. Phys., 11(6), 2689–2701, doi:10.5194/acp-11-2689-2011, 2011.

Eresmaa, N., Härkönen, J., Joffre, S. M., Schultz, D. M., Karppinen, A. and Kukkonen, J.: A Three-Step Method for Estimating the Mixing Height Using Ceilometer Data from the Helsinki Testbed, J. Appl. Meteorol. Climatol., 51(12), 2172–2187, doi:10.1175/JAMC-D-12-058.1, 2012.

Gregorio, E., Rocadenbosch, F. and Comeron, A.: Design methodology of a ceilometer lidar prototype, in 2007 IEEE International Geoscience and Remote Sensing Symposium, pp. 3162–3165, IEEE, 2007.

Haeffelin, M., Barthès, L., Bock, O., Boitel, C., Bony, S., Bouniol, D., Chepfer, H., Chiriaco, M., Cuesta, J., Delanoë, J., Drobinski, P., Dufresne, J.-L., Flamant, C., Grall, M., Hodzic, A., Hourdin, F., Lapouge, F., Lemaître, Y., Mathieu, A., Morille, Y., Naud, C., Noël, V., O'Hirok, W., Pelon, J., Pietras, C., Protat, A., Romand, B., Scialom, G. and Vautard, R.: SIRTA, a ground-based atmospheric observatory for cloud and aerosol research, Ann. Geophys., 23(2), 253–275, doi:10.5194/angeo-23-253-2005, 2005.

Haeffelin, M., Bergot, T., Elias, T., Tardif, R., Carrer, D., Chazette, P., Colomb, M., Drobinski, P., Dupont, E., Dupont, J.-C., Gomes, L., Musson-Genon, L., Pietras, C., Plana-Fattori, A., Protat, A., Rangognio, J., Raut, J.-C., Rémy, S., Richard, D., Sciare, J. and Zhang, X.: PARISFOG: Shedding New Light on Fog Physical Processes, Bull. Am. Meteorol. Soc., 91(6), 767–783, doi:10.1175/2009BAMS2671.1, 2010.

Haeffelin, M., Angelini, F., Morille, Y., Martucci, G., Frey, S., Gobbi, G. P., Lolli, S., O'Dowd, C. D., Sauvage, L., Xueref-Rémy, I., Wastine, B. and Feist, D. G.: Evaluation of Mixing-Height Retrievals from Automatic Profiling Lidars and Ceilometers in View of Future Integrated Networks in Europe, Boundary-Layer Meteorol., 143(1), 49–75, doi:10.1007/s10546-011-9643-z, 2012.

Heese, B., Flentje, H., Althausen, D., Ansmann, A. and Frey, S.: Ceilometer lidar comparison: backscatter coefficient retrieval and signal-to-noise ratio determination, Atmos. Meas. Tech., 3(6), 1763–1770, doi:10.5194/amt-3-1763-2010, 2010.

Hervo, M., Poltera, Y. and Haefele, A.: An empirical method to correct for temperature dependent variations in the overlap function of CHM15k ceilometers, Atmos. Meas. Tech. Discuss., 1–27, doi:10.5194/amt-2016-30, 2016.

Hopkin, E., Illingworth, A., Westbrook, C., Charlton-Perez, C., and Ballard, S.: Calibration of the Met Office Network using the Cloud Method, ISARS, Varna, Bulgaria, 6-9 June 2016.

Illingworth, A. J., Hogan, R. J., O'Connor, E. J., Bouniol, D., Delanoë, J., Pelon, J., Protat, A., Brooks, M. E., Gaussiat, N., Wilson, D. R., Donovan, D. P., Baltink, H. K., van Zadelhoff, G.-J., Eastment, J. D., Goddard, J. W. F., Wrench, C. L., Haeffelin, M., Krasnov, O. A., Russchenberg, H. W. J., Piriou, J.-M., Vinit, F., Seifert, A., Tompkins, A. M. and Willén, U.: Cloudnet, Bull. Am. Meteorol. Soc., 88(6), 883–898, doi:10.1175/BAMS-88-6-883, 2007.

Illingworth, A. J., Cimini, D., Gaffard, C., Haeffelin, M., Lehmann, V., Löhnert, U., O'Connor, E. J. and Ruffieux, D.: Exploiting existing ground-based remote sensing networks to improve high resolution weather forecasts, Bull. Am. Meteorol. Soc., 96, 2107–2125, doi:10.1175/BAMS-D-13-00283.1, 2015.

Knippertz, P. and Stuut, J.-B. W.: Mineral Dust: A Key Player in the Earth System, edited by P. Knippertz and J.-B. W. Stuut, Springer Netherlands., 2014.

Madonna, F., Amato, F., Vande Hey, J. and Pappalardo, G.: Ceilometer aerosol profiling versus Raman lidar in the frame of the INTERACT campaign of ACTRIS, Atmos. Meas. Tech., 8(5), 2207–2223, doi:10.5194/amt-8-2207-2015, 2015.

Manninen, A. J., O'Connor, E. J., Vakkari, V. and Petäjä, T.: A generalised background correction algorithm for a Halo Doppler lidar and its application to data from Finland, Atmos. Meas. Tech. Discuss., 8(10), 11139–11170, doi:10.5194/amtd-8-11139-2015, 2015.

Markowicz, K. M., Flatau, P. J., Kardas, A. E., Remiszewska, J., Stelmaszczyk, K. and Woeste, L.: Ceilometer Retrieval of the Boundary Layer Vertical Aerosol Extinction Structure, J. Atmos. Ocean. Technol., 25(6), 928–944, doi:10.1175/2007JTECHA1016.1, 2008.

Martucci, G., Milroy, C. and O'Dowd, C. D.: Detection of Cloud-Base Height Using Jenoptik CHM15K and Vaisala CL31 Ceilometers, J. Atmos. Ocean. Technol., 27(2), 305–318, doi:10.1175/2009JTECHA1326.1, 2010.

McKendry, I. G., van der Kamp, D., Strawbridge, K. B., Christen, A. and Crawford, B.: Simultaneous observations of boundary-layer aerosol layers with CL31 ceilometer and 1064/532 nm lidar, Atmos. Environ., 43(36), 5847–5852, doi:10.1016/j.atmosenv.2009.07.063, 2009.

Mielonen, T., Aaltonen, V., Lihavainen, H., Hyvärinen, A.-P., Arola, A., Komppula, M. and Kivi, R.: Biomass Burning Aerosols Observed in Northern Finland during the 2010 Wildfires in Russia, Atmosphere (Basel)., 4(1), 17–34, doi:10.3390/atmos4010017, 2013.

Münkel, C., Eresmaa, N., Räsänen, J. and Karppinen, A.: Retrieval of mixing height and dust concentration with lidar ceilometer, Boundary-Layer Meteorol., 124(1), 117–128, doi:10.1007/s10546-006-9103-3, 2006 2007.

Münkel, C., Emeis, S., Schäfer, K. and Brümmer, B.: Improved near-range performance of a low-cost one lens lidar scanning the boundary layer, Proc. SPIE, 7475, 1–10, doi:10.1117/12.830397, 2009.

Nemuc, A., Stachlewska, I. S., Valilescu, J., Górska, A., Nicolae, D. and Talianu, C.: Optical Properties of Long-Range Transported Volcanic Ash over Romania and Poland During Eyjafjallajökull Eruption in 2010, Acta Geophys. 62, 350-366 doi: 10.2478/s11600-013-0180-7, 2014.

O'Connor, E. J., Illingworth, A. J. and Hogan, R. J.: A Technique for Autocalibration of Cloud Lidar, J. Atmos. Ocean. Technol., 21(5), 777–786, doi:10.1175/1520-0426(2004)021<0777:ATFAOC>2.0.CO;2, 2004.

O'Connor, E. J., Hogan, R. J. and Illingworth, A. J.: Retrieving Stratocumulus Drizzle Parameters Using Doppler Radar and Lidar, J. Appl. Meteorol., 44(1), 14–27, doi:10.1175/JAM-2181.1, 2005.

Rogers, R. R., Lamoureux, M.-F., Bissonnette, L. R. and Peters, R. M.: Quantitative Interpretation of Laser Ceilometer Intensity Profiles, J. Atmos. Ocean. Technol., 14(3), 396–411, doi:10.1175/1520-0426(1997)014<0396:QIOLCI>2.0.CO;2, 1997.

Selvaratnam, V., Ordnez, C. and Adam, M.: Comparison of planetary boundary layer heights from Jenoptik ceilometers and the Unified Model, Forecast. Res. Tech. Rep. No 605, 24, 2015.

Sokół, P., Stachlewska, I., Ungureanu, I. and Stefan, S.: Evaluation of the boundary layer morning transition using the CL-31 ceilometer signals, Acta Geophys., 62(2), 367–380, doi:10.2478/s11600-013-0158-5, 2014.

Stachlewska, I. S., Piądłowski, M., Migacz, S., Szkop, A., Zielińska, A. J. and Swaczyna, P. L.: Ceilometer observations of the boundary layer over Warsaw, Poland, Acta Geophys., 60(5), 1386–1412, doi:10.2478/s11600-012-0054-4, 2012.

Sundström, A.-M., Nousiainen, T. and Petäjä, T.: On the Quantitative Low-Level Aerosol Measurements Using Ceilometer-Type Lidar, J. Atmos. Ocean. Technol., 26(11), 2340–2352, doi:10.1175/2009JTECHA1252.1, 2009.

Tang, G., Zhang, J., Zhu, X., Song, T., Münkel, C., Hu, B., Schäfer, K., Liu, Z., Zhang, J., Wang, L., Xin, J., Suppan, P. and Wang, Y.: Mixing layer height and its implications for air pollution over Beijing, China, Atmos. Chem. Phys., 16(4), 2459–2475, doi:10.5194/acp-16-2459-2016, 2016.

Tsaknakis, G., Papayannis, A., Kokkalis, P., Amiridis, V., Kambezidis, H. D., Mamouri, R. E., Georgoussis, G. and Avdikos, G.: Inter-comparison of lidar and ceilometer retrievals for aerosol and Planetary Boundary Layer profiling over Athens, Greece, Atmos. Meas. Tech., 4(6), 1261–1273, doi:10.5194/amt-4-1261-2011, 2011.

Vande Hey, J. D.: A Novel Lidar Ceilometer: Design, Implementation and Characterisation, Springer, 2014.

van der Kamp, D.: Ceilometer observations of Vancouver's urban boundary layer: validation and mixed-layer height estimation, Master's thesis, University of British Columbia, doi:10.14288/1.0066572, 2008.

van der Kamp, D. and McKendry, I.: Diurnal and Seasonal Trends in Convective Mixed-Layer Heights Estimated from Two Years of Continuous Ceilometer Observations in Vancouver, BC, Boundary-Layer Meteorol., 137(3), 459–475, doi:10.1007/s10546-010-9535-7, 2010.

Weitkamp, C.: Lidar: Range-Resolved Optical Remote Sensing of the Atmosphere, Springer, New York, 2005.

Welch, B. L.: The generalization of "Student's" problem when several different population variances are involved, Biometrika, 34, 28–35, doi:10.1093/biomet/34.1-2.28, 1947.

Wiegner, M. and Gasteiger, J.: Correction of water vapor absorption for aerosol remote sensing with ceilometers, Atmos. Meas. Tech., 8(9), 3971–3984, doi:10.5194/amt-8-3971-2015, 2015.

Wiegner, M., Gasteiger, J., Groß, S., Schnell, F., Freudenthaler, V., Forkel, R., 2012. Characterization of the Eyjafjallajökull ash-plume: Potential of lidar remote sensing. Phys. Chem. Earth, 45-46, 79–86. doi:10.1016/j.pce.2011.01.006

Wiegner, M., Madonna, F., Binietoglou, I., Forkel, R., Gasteiger, J., Geiß, A., Pappalardo, G., Schäfer, K. and Thomas, W.: What is the benefit of ceilometers for aerosol remote sensing? An answer from EARLINET, Atmos. Meas. Tech., 7(7), 1979–1997, doi:10.5194/amt-7-1979-2014, 2014.

Xie, C. and Zhou, J.: Method and analysis of calculating signal-to-noise ratio in lidar sensing, Spie, Proc. SPIE 5832, Opt. Technol. Atmos. Ocean. Environ. Stud., 738, doi:10.1117/12.619881, 2005.

Table 1: Internal averaging interval applied in different CL31 firmware versions as a function of range *r* and reporting interval.

|  |  | Reporting interval | | | |
|---|---|---|---|---|---|
| Firmware version | Range [m] | 2 s | 3 – 4 s | 5 – 8 s | > 8 s |
| <1.72, 2.01, 2.02 | $r < 600$ | 2 s | 4 s | 8 s | 16 s |
|  | $600 \leq r < 1200$ | 4 s | 4 s | 8 s | 16 s |
|  | $1200 \leq r < 1800$ | 8 s | 8 s | 8 s | 16 s |
|  | $1800 \leq r < 2400$ | 16 s | 16 s | 16 s | 16 s |
|  | $r \geq 2400$ | 30 s | 30 s | 30 s | 30 s |
| 1.72, 2.03 | $r > 0$ | 30 s | 30 s | 30 s | 30 s |

Table 2: Vaisala CL31 ceilometer specifications of sensor hardware, firmware, noise setting and resolution applied by the user. The term 'H2' is discussed in Sect. 3.2. *Block averages of the recorded data (2 s, 5 m) are used for sensor S.

| Sensor ID | Network | Ceilometer Engine Board / Receiver / Transmitter | Firmware version | H2 | Resolution (time, range) |
|---|---|---|---|---|---|
| A | LUMO | CLE311 / CLE311 / CLT311 | 1.56, 1.61, 1.71 | On | 15 s, 10 m |
|  |  | CLE311 / CLE311 / CLT321 | 1.71, 1.72 | On | 15 s, 10 m |
| B | LUMO | CLE311 / CLE311 / CLT311 | 1.61 | On | 15 s, 10 m |
|  |  | CLE311 / CLE311 / CLT321 | 1.61, 1.71, 1.72 | On | 15 s, 10 m |
| C | LUMO | CLE321 / CLE321 / CLT321 | 2.01, 2.02, 2.03 | On | 15 s, 10 m |
| D | LUMO | CLE321 / CLE321 / CLT321 | 2.01, 2.02, 2.03 | On | 15 s, 10 m |
| W | Met Office | CLE311 / CLE311 / CLT311 | 1.71 | Off | 30 s, 20 m |
| S | Meteo France | CLE321 / CLE321 / CLT321 | 2.01 | On | 30 s, 15 m* |

Table 3: Instrument specific correction function coefficients to address systematic alterations in the lowest 100 m introduced by a hardware related perturbation for four LUMO sensors (Table 2) in 2013 when operating with firmware versions 1.61 (A & B) and 2.01 (C & D), respectively. The intercept b and slope a are given for a linear regression by range gate: $\tilde{P}^{OC}(n)/\tilde{P}^{OC}(10) = a \cdot \tilde{P}^{OC}(5)/\tilde{P}^{OC}(10) + b$ with range gate $n \in (6, 7, 8, 9)$.

| Range gate | b (intercept) | | | | a (slope) | | | |
|---|---|---|---|---|---|---|---|---|
| | A | B | C | D | A | B | C | D |
| 6 | 1.4190 | 0.3557 | 0.0210 | -0.6552 | -0.1654 | 0.5656 | 0.8651 | 1.3990 |
| 7 | 0.8661 | 0.4197 | 0.6530 | 0.4471 | 0.1648 | 0.5099 | 0.3023 | 0.4734 |
| 8 | 0.7581 | 0.5638 | 0.8765 | 0.9456 | 0.1649 | 0.3349 | 0.0613 | 0.0147 |
| 9 | 0.8367 | 0.7739 | 0.9634 | 1.0057 | 0.1111 | 0.1830 | 0.0187 | -0.0107 |

[Figure]

Fig. 1: Range histograms for 24 h of observations on different clear-sky days from Vaisala CL31 ceilometers operating with firmware versions (1.56 – 2.03). Sensor ID in brackets (see Table 2 for settings, e.g. H2  off for sensor W). Rows: range histograms in arbitrary units (a.u.) of (a)   signal $\tilde{P}^{raw}$,$P$, (b) range-corrected  signal $\tilde{P}^{raw}$ reported $RCS = P \cdot r^2$, and (c) range-corrected,  background-corrected signal $\tilde{P}\hat{P} \cdot r^2$ (Eq. (5)). Median profiles (solid lines) are included in (b) and (c). The H2 setting  (Sect. 3.2  allows switchoff of the range correction above 2400 m for regions with no clouds present.

[Figure]

Fig. 2: Signal *P* (derived from reported signal $P^{raw}$ (by reverting range correction) observed with  Vaisala CL31 sensors operating with (a, b) engine board CLE311 + receiver CLR311 (A , B) and (c, d, e) CLE321 + CLR321 (C , D, S), respectively  (Table 2) (a-d) January 2011  April 2016 and (e) May 2015 – April 2016 for range > 2400 . Observations (four hours around midnight, 22-02 UTC) are hourly means of profiles when: clouds detected for < 10% of the hour, no fog, average window transmission > 80%, laser pulse energy > 98% and data availability > 90%. (left) Top axis shows firmware updates (version 1.71, then 1.72 for sensors A & B; versions 2.02, then 2.03 for C & D; version 2.01 for S) and hardware changes/upgrades (transmitter CLT311 replaced by CLT321 for sensors A and B; CLT321 replaced by a new CLT321 for sensor S). (right) median profiles (with IQR shading) of all selected observations grouped by firmware version and transmitter , with N indicating the number of profiles.

[Figure]

Fig. 3: Long-term median vertical profiles of range-dependent  background  $P^B$ for Vaisala CL31 sensors (Table 2). Statistics are based on hourly mean profiles > 2410  m) of reported signal with after reverting the range-correction $P^{reported}$ observed around midnight (same data as Fig. 2).  Ceilometers A & B operated with firmware 1.61, 1.71 or 1.72 and transmitter type CLT311 or CLT321, respectively; ceilometers C & D operated with CLT321 and firmware 2.01, 2.02, and 2.03; ceilometer S operated with firmware 2.01 and CLT321. (a) Median profiles for each sensor calculated separately by firmware version for sensors A & B, all are combined for C & D (2.xx) due to their similarity; (b) as in (a) for sensors A, C, B, D but also separating by ceilometer transmitter CLT and laser heat sink temperature combinations (see legend); laser heat sink temperature (as reported by the ceilometer) is used to subdivide profiles into three classes ($T_{laser} < 303$, $303 \leq T_{laser} < 308$ K, and $T_{laser} \geq 308$ K.), (c) as in (b) but for selected profiles (solid lines, A & B with 1.71 and 1.72; C & D with 2.03) and their respective background profiles as determined by a 30-min termination hood measurement at the same setting and laser heat sink temperature class (thick lines); (d) as in (c) but range-corrected; and (e) as in (d) but zoomed into the range < 3000 m. Number of hourly mean profiles N [h] available for each combination of sensor, firmware, transmitter type CLT and laser temperature is listed in the legend. Profiles are smoothed vertically with a moving average over a window of ~~(A, C, D) 210 m and (B) 310 m, given the wave-type bias did not average out sufficiently for sensor B. (a) All available climatological profiles; (b) Selected climatological profiles (solid lines) and their respective background profiles as determined by a 30 min termination hood measurement at the same setting and laser heat sink temperature class (thick lines): sensors operating with CLT321 and firmware 1.71 and 1.72 (sensors A & B) and firmware 2.03 (sensors C & D).~~210 m, only for profiles from sensor B a smoothing window of 310 m is used.

[Figure]

Fig. 4 : Logarithm of range-corrected signal reported $RCS = P \cdot r^2$ in the first five range gates during a termination hood measurement of LUMO sensor A with firmware 1.72.

[Figure]

Fig. 5: Manufacturer-deduced overlap function of Vaisala CL31 ceilometers using firmware versions 1.71, 1.72, 2.02, or 2.03 (older versions used an overlap function with 5 % to 10 % lower overlap values). The function, applied in the lowest range gates above the instrument, is derived from laboratory measurements and field observations under homogeneous atmospheric conditions. During the production process, the applicability of the overlap function is verified for each unit. Due to the stable instrument conditions (e.g. low internal temperature variations), Vaisala expects no systematic variations of the overlap function. The error is stated to be below 10%.

[Figure]

Fig. 6: Median  range-corrected signal $\tilde{P}^{OC}$ reported $RCS = P \cdot r^2$ of the lowest 9 range gates (10 – 90 m) normalised by the value at the $10^{th}$ range gate for four LUMO sensors (Table 2) with firmware versions (a-c) 1.61 (A, B) or 2.01 (C, D) in 2013 and (d) 1.72 (A, B) or 2.03 (C, D) in 2015-2016, respectively. Statistics calculated for all profiles observed between 11-16 UTC with $\tilde{P}^{OC}$ $RCS < 200 \times 10^{-8}$ a.u. in the lowest 400 m: median (solid line) and inter-quartile rage (shading). Panels (b) and (c) separate the profiles from panel (a): into (b) profiles with the ratio at the $3^{rd}$ range gate, i.e. $(|RCS(3)/RCS($10$)|$ exceeding or equal to 0.8, while (c) shows the profiles with the same ratio less than 0.8. For (a, d) the total number of 15 s profiles selected is indicated by sensor A, B, C or D and in (b, c) the percentages of the values from the total number of profiles in panel (a) are given.

[Figure]

Fig. 7: Observations from four  LUMO network (Table 2) over the first 200 m range: (i, iii, v, vii) logarithm of the  range-corrected  signal reported *RCS* [a.u.] and (ii, iv, vi, viii) as in (i, iii, v, vii), but after application of correction for near-range artefacts associated with the obstruction correction and a hardware-related perturbation (see Fig. 6). Sensors were operating with firmware 1.61 (A, B) and 2.01 (C, D) on (i, ii) 10/01/2014, (iii, iv) 15/01/ 2014, (v, vi) 06/01/2013, and firmware 1.72 and 2.03 on (vii, viii) 13/03/2016. White colours indicate values outside of the range of values selected (see colour legends). Note that data are not absolutely calibrated.

[Figure]

Fig. 8: Observations from Vaisala CL31 sensor D (Table 2) at top range gates (7410 – 7700 m) with cirrus during the early evening on 1 February 2013: (a) relative variance $RV$ (Eq. (14)), (b)  background-corrected signal $\hat{P}$, (c) same as (b), but only including observations with $RV > 1$, and (d) time series of the noise floor $F$ (Eq. (13)) based on the cleaned signal shown in (c) with missing values interpolated linearly.

[Figure]

Fig. 9: Acceptance [%] based on Welch's t-test with a p-value of 0.01 of smoothed, not range corrected attenuated backscatter (Eq. (12)) to be significantly higher than the noise floor (Eq. (14)), binned by the corresponding signal-­to-­noise ratio (*SNR*, Eq. 14) for four selected cases (24 h observations; range 50 – 300 m shown for simplicity) of observations taken with different firmware versions (see legend). The shaded area marks the *SNR* region corresponding to acceptance levels of 50 – 90%.

[Figure]

Fig. 10: Logarithm of range corrected attenuated backscatter β for the lowest 3000 m for a clear-sky day (: CL31 observations on 24 July 2012: (row 1 & 2) and a day with boundary layer clouds (), 22 June 2014: (row 3 & 4), and 29 June 2016 (row 5) from three CL31 ceilometers (A four sensors with firmware 1.61, B with firmware 1.71, and C with version in brackets: A (1.61), B (1.71), C (2.02,), and S (20.1), see Table 2):. (a) Range-corrected attenuated backscatter at recording interval of 15 s without correction for cosmetic shift resolution (row 1–4) and electronic background (see Sect. 3.1),30 s (row 5) as reported, (b) as in (a) with running average (101 time steps, 11 range gates(~ 25 min, ~ 100 m) applied (Eq. (12)), (c) as in (b) but for attenuated backscatter from entirelyincluding correction of instrument-related background corrected signal,

and potential cosmetic shift (see Sect. 3.1), and (d) as in (c) but filtered for $SNR > T_2$, with $T_2 = 0.18$ (see discussion on Fig. 9). Note: for simplicity the absolute calibration constant is here assumed to be $c_{absolute} = 1$ (Sect. 4.1) for all sensors. This is not necessarily expected to be a correct assumption in reality but applied to show the impact of corrections on the final product, i.e. the attenuated backscatter.